# Long-range projection neurons in the taste circuit of *Drosophila*

**Heesoo Kim[1,2]\*, Colleen Kirkhart[1,2], Kristin Scott[1,2]\***

[1]Department of Molecular and Cell Biology, University of California, Berkeley, Berkeley, United States; [2]Helen Wills Neuroscience Institute, University of California, Berkeley, Berkeley, United States

**Abstract** Taste compounds elicit innate feeding behaviors and act as rewards or punishments to entrain other cues. The neural pathways by which taste compounds influence innate and learned behaviors have not been resolved. Here, we identify three classes of taste projection neurons (TPNs) in *Drosophila melanogaster* distinguished by their morphology and taste selectivity. TPNs receive input from gustatory receptor neurons and respond selectively to sweet or bitter stimuli, demonstrating segregated processing of different taste modalities. Activation of TPNs influences innate feeding behavior, whereas inhibition has little effect, suggesting parallel pathways. Moreover, two TPN classes are absolutely required for conditioned taste aversion, a learned behavior. The TPNs essential for conditioned aversion project to the superior lateral protocerebrum (SLP) and convey taste information to mushroom body learning centers. These studies identify taste pathways from sensory detection to higher brain that influence innate behavior and are essential for learned responses to taste compounds.

## Introduction

The ability to detect nutrients and toxins is critical for survival. In most animals, the gustatory system has evolved as a specialized sensory system to detect non-volatile chemicals in the environment. The taste system innately drives strong food acceptance or rejection behaviors. In addition, gustatory detection of nutrients can serve as a reward to impart positive valence to previously neutral stimuli whereas bitter detection can act as a punishment signal. The pathways by which gustatory detection drive innate feeding behavior and learned associations have not been resolved.

*Drosophila melanogaster*, like other animals, uses the gustatory system to detect nutrients and toxins in potential food. Fly gustatory receptor neurons (GRNs) are located on the proboscis labellum, internal mouthparts, legs, and wings (*Stocker, 1994*). Different GRNs detect different taste modalities, including, sugar, bitter, water, and pheromones (*Cameron et al., 2010*; *Chen et al., 2010*; *Liman et al., 2014*; *Lu et al., 2012*; *Thistle et al., 2012*; *Thorne et al., 2004*; *Toda et al., 2012*; *Wang et al., 2004*). Most sensory neurons on the legs and wings send axons to the ventral nerve cord (VNC), whereas those on the proboscis labellum and mouthparts project to the subesophageal zone (SEZ) of the central brain (*Stocker, 1994*; *Wang et al., 2004*). Activation of sucrose-responsive taste neurons triggers behavioral subprograms associated with feeding, including inhibition of locomotion, extension of the fly proboscis, and ingestion (*Gordon and Scott, 2009*; *Keene and Masek, 2012*; *Marella et al., 2006*; *Thoma et al., 2016*). In contrast, activation of bitter-responsive sensory neurons triggers food rejection, including proboscis retraction (*Keene and Masek, 2012*; *Marella et al., 2006*). In addition, taste cell activation elicits responses in mushroom body learning circuits for sensory associations (*Burke et al., 2012*; *Kirkhart and Scott, 2015*; *Liu et al., 2012*).

**\*For correspondence:** heesoo@ berkeley.edu (HK); kscott@ berkeley.edu (KS)

Second-order neurons that respond to taste stimuli and transmit this information for innate and learned behaviors have been challenging to identify. A recent study implicated a pair of neurons that projects from the SEZ to the antennal mechanosensory and motor center (AMMC) as candidate second-order taste neurons. These neurons respond to sugar stimulation of proboscis taste neurons and promote proboscis extension (*Kain and Dahanukar, 2015*). A different study identified local SEZ interneurons as candidate second-order neurons that sense sugar detection specifically of mouthpart sensory neurons. Inhibiting activity in these neurons selectively inhibited ingestion, and activating them prolonged ingestion without influencing proboscis extension (*Yapici et al., 2016*). The candidate second-order neurons identified to date show remarkable specificity in terms of the sensory neurons that activate them (proboscis versus mouthparts) and the behavioral subprograms they generate (proboscis extension versus ingestion). The model suggested by these findings is that taste information is processed by parallel labeled lines via several different neural streams that coordinate different aspects of feeding behavior. Consistent with this, a behavioral study of the function of different taste neurons on the legs found that some caused inhibition of locomotion whereas others promoted proboscis extension (*Thoma et al., 2016*).

To test the specialization of taste processing streams beyond the sensory neurons, we searched for candidate second-order neurons that might convey taste information from the legs or the proboscis to the higher brain. Here, we identify three classes of taste projection neurons (TPNs), distinguished by their morphology and taste selectivity, that reveal the existence of taste processing streams that are especially important for learned associations. Our studies demonstrate the existence of modality-selective taste pathways to higher brain, highlight similarities in the organization of second-order sensory neurons, and characterize taste processing streams required for learned taste aversion.

## Results

### Identification of three candidate taste projection neurons

GRNs on the legs project to the leg ganglia of the VNC whereas those from the proboscis labellum and mouthparts project to the SEZ (*Figure 1A*; *Table 1* contains genotypes for all experiments). To identify novel second-order taste projection neurons (TPNs), we searched for neurons with arborizations near GRN axons (in the VNC and SEZ) in a visual screen of more than 8000 images of Gal4 lines from existing collections (Dickson, unpublished; *Gohl et al., 2011*; *Jenett et al., 2012*). We identified three candidate TPN classes labeled by sparse Gal4 lines, which we name TPN1, TPN2, and TPN3.

TPN1, marked by *R30A08-Gal4,* has two bilaterally symmetric cell bodies in the metathoracic (hind leg) ganglia, contralateral dendrites in each leg ganglion, and axons that project from the VNC to terminate in the SEZ (*Figure 1B* and *Figure 1—figure supplement 1A,B*). TPN2 contains two pairs of bilaterally symmetric cells, labeled by *VT57358-Gal4*. One pair has cell bodies on the dorsal surface of the metathoracic neuromere with contralateral VNC dendrites and the second pair has cell bodies in the abdominal ganglion with bilateral VNC dendrites (*Figure 1C* and *Figure 1—figure supplement 1C*). Otherwise, the two pairs have similar axonal projections in the SEZ and lateral protocerebrum (*Figure 1—figure supplement 1D,E*). TPN3 has one pair of cell bodies in the cervical connective, bilateral dendrites in the SEZ, and bilateral axonal termini in the lateral protocerebrum (*Figure 1D* and *Figure 1—figure supplement 1I*). We were unable to find a driver line specific for TPN3, but found two broader Gal4 lines, *C220-Gal4* and *R11H09-Gal4,* that contain TPN3 in addition to non-overlapping neurons (*Figure 1—figure supplement 1F–H*). Thus, the three TPN classes that we identified have arbors in brain regions near taste sensory projections from the legs and proboscis and terminate either in the SEZ (which contains other taste-responsive cells) or in the higher brain.

### TPNs dendrites are in close proximity to gustatory receptor neuron axons

To test the possibility that candidate TPNs directly receive signals from gustatory neurons, we investigated the proximity of TPN dendrites to GRN axons. We expressed the CD8-tdTomato red fluorescent reporter in each of the TPNs while simultaneously expressing the CD2-GFP green fluorescent

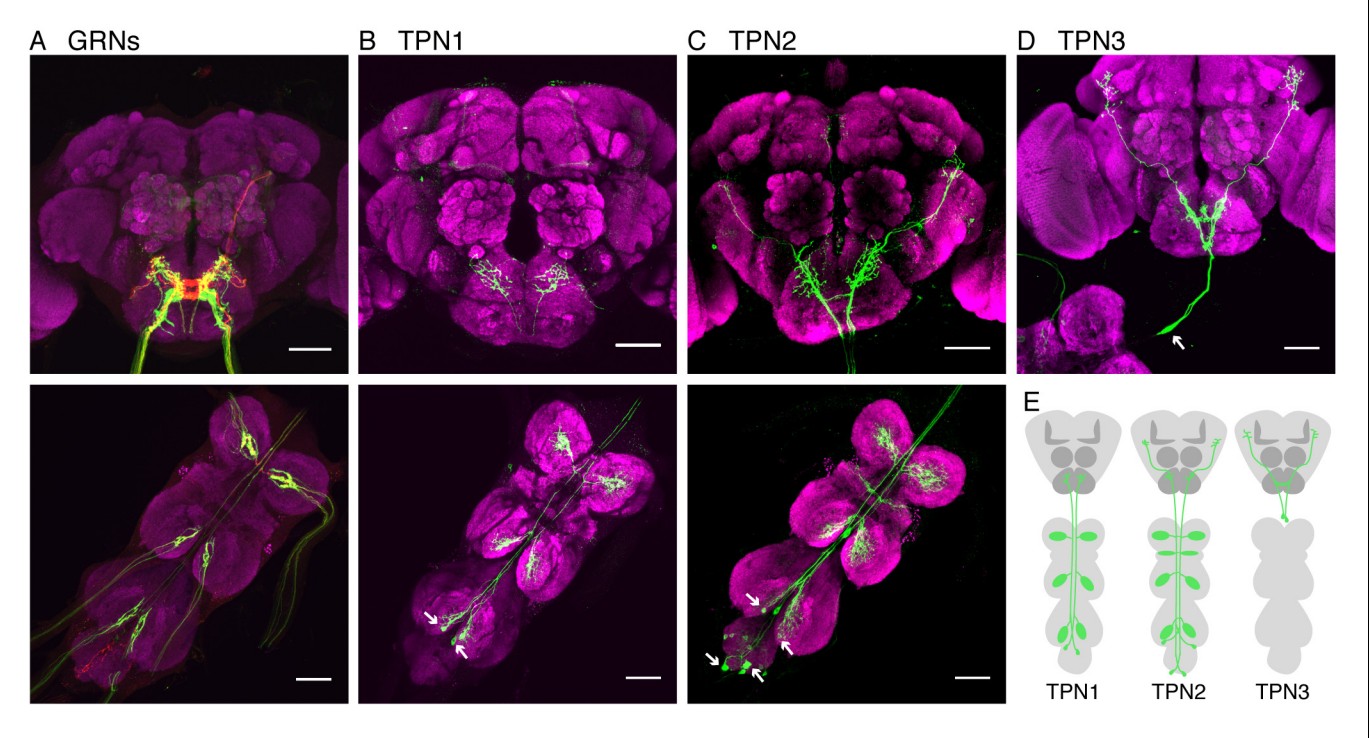

**Figure 1.** Identification of taste projection neurons. (**A**) Projection pattern of bitter (red) and sugar (green) GRNs into the SEZ (top) and VNC (bottom). For A-D, magenta indicates neuropil; arrows indicate cell bodies; and scale bar, 50 µm. (**B**) Full expression of *R30A08-Gal4*, showing TPN1 fibers (green) in SEZ (top) and cell bodies and fibers in the VNC (bottom). (**C**) Full expression of *VT57358-Gal4*, showing TPN2 fibers (green) in SEZ and protocerebrum (top) and cell bodies and fibers in the VNC (bottom). (**D**) Single-cell clone of *C220-Gal4*, showing anatomy of TPN3 (green). (**E**) Schematic of anatomical patterns of the taste projection neurons.

The following figure supplement is available for figure 1:

**Figure supplement 1.** Detailed anatomy of TPNs.

reporter in one of the four major GRN classes (sweet, bitter, water or pheromone). We found that the dendrites of TPN1 and TPN2 overlapped specifically with sweet GRN axons in the VNC, but not bitter, pheromone, or water GRN axons. Conversely, the dendrites of TPN3 overlapped specifically with bitter GRNs in the SEZ, but not sugar, pheromone or water GRN axons (*Figure 2A* and *Figure 2—figure supplement 1*).

As a further test of proximity, we performed GFP Reconstitution Across Synaptic Partners (GRASP) experiments in which a membrane-tethered split GFP was expressed in TPNs and the complementary membrane-tethered split GFP in GRNs, such that GFP would be reconstituted only if TPNs and GRNs were adjacent (*Feinberg et al., 2008*; *Gordon and Scott, 2009*) (*Figure 2B,C*). There was significant GRASP signal between TPN1/TPN2 and sugar GRNs in the VNC, suggesting a direct connection between leg sugar GRNs and both TPN1 and TPN2. In addition, strong GRASP signal was found between TPN3 and bitter GRNs in the SEZ, both in the anterior ring-like structure (from the labellar bitter GRNs) and the posterior V-like structure (from the leg bitter GRNs) (*Wang et al., 2004*). Limited signal was detected between TPN1/TPN2 and bitter GRNs or TPN3 and sugar GRNs, arguing for modality-specific connections between GRNs and TPNs. These anatomical studies suggest that TPN1 and TPN2 may convey sugar detection on the legs to the SEZ or higher brain whereas TPN3 may convey bitter signals from the legs and proboscis to higher brain.

## TPNs respond to taste compounds

Our anatomical studies suggest that TPN1 and TPN2 are well positioned to respond to sugar stimulation and TPN3 is well positioned to respond to bitter stimulation. To directly test if the TPNs are

**Table 1.** Genotypes of Experimental Flies

| Figure | Description | Genotype |
|---|---|---|
| 1A | GRN anatomy | Gr66a-Gal4/CyO; UAS-cd8-tdTomato/Gr64fLexA, LexAop-cd2-GFP |
| 1B | TPN1 anatomy | UAS-GFP/+; R30A08-Gal4/+ |
| 1C | TPN2 anatomy | UAS-GFP/+; VT57358-Gal4/+ |
| 1D | TPN3 anatomy | tub>Gal80>/X; UAS-GFP/+; C220-Gal4/hsFLP, MKRS |
| 1 – fs1A | TPN1 single | UAS-GFP/+; R30A08-Gal4/+ |
| 1 - fs1B | TPN1 dendrite/axon | +/CyO; R30A08-Gal4/UAS-DenMark, UAS-synaptotagmin-eGFP |
| 1 - fs1C & D | TPN2 mosaics | tub>Gal80>/X; UAS-GFP/+; VT57358-Gal4/ hsFLP, MKRS |
| 1 - fs1E | TPN2 dendrite/axon | +/CyO; VT57358-Gal4/UAS-DenMark, UAS-synaptotagmin-eGFP |
| 1 - fs1F | TPN3 – 2nd Gal4 | UAS-GFP/+; C220-Gal4/+ |
| 1 - fs1G | TPN3 – 1st Gal4 | UAS-GFP/+; R11H09-Gal4/+ |
| 1 - fs1H | TPN3 – 2 cells only | UAS-GFP/UAS-RedStinger; C220-Gal4/R11H09-Gal4 |
| 1 - fs1I | TPN3 dendrite/axon | +/CyO; C220-Gal4/UAS-DenMark, UAS-synaptotagmin-eGFP |
| 2A & 2 – fs1 | Sugar GRN doubles | UAS-cd8tdTomato/CyO; R30A08-Gal4 / Gr64fLexA, LexAop-cd2-GFP<br>UAS-cd8tdTomato/CyO; VT57358-Gal4 / Gr64fLexA, LexAop-cd2-GFP<br>UAS-cd8tdTomato/CyO; C220-Gal4 / Gr64fLexA, LexAop-cd2-GFP |
| 2A & 2 – fs1 | Bitter GRN doubles | UAS-cd8tdTomato/Gr66a-LexA; R30A08-Gal4 / LexAop-cd2-GFP<br>UAS-cd8tdTomato/Gr66a-LexA; VT57358-Gal4 / LexAop-cd2-GFP<br>UAS-cd8tdTomato/Gr66a-LexA; C220-Gal4 / LexAop-cd2-GFP |
| 2A | Pheromone GRN doubles | UAS-cd8tdTomato/ppk23-LexA; R30A08-Gal4 / LexAop-cd2-GFP<br>UAS-cd8tdTomato/ppk23-LexA; VT57358-Gal4 / LexAop-cd2-GFP<br>UAS-cd8tdTomato/ppk23-LexA; C220-Gal4 / LexAop-cd2-GFP |
| 2A | Water GRN doubles | UAS-cd8tdTomato/ppk28-LexA, LexAop-cd2-GFP; R30A08-Gal4 / TM2 or 6b<br>UAS-cd8tdTomato/ppk28-LexA, LexAop-cd2-GFP; VT57358-Gal4 / TM2 or 6b<br>UAS-cd8tdTomato/ppk28-LexA, LexAop-cd2-GFP; C220-Gal4 / TM2 or 6b |
| 2B | Sugar GRN GRASP | Gr5a-LexA/X; UAS-cd8tdTomato/LexAop-CD4::spGFP11;<br>R30A08-Gal4 or VT57358-Gal4 or C220-Gal4/UAS-CD4::spGFP1-10 |
| 2B | Bitter GRN GRASP | Gr66a-LexA/X; UAS-cd8tdTomato/LexAop-CD4::spGFP11;<br>R30A08-Gal4 or VT57358-Gal4 or C220-Gal4/UAS-CD4::spGFP1-10 |
| 3 A-F top | TPN1 GCaMP | UAS-cd8tdTomato/UAS-GCaMP6s; R30A08-Gal4/UAS-GCaMP6s |
| 3 A-F middle | TPN2 GCaMP | UAS-cd8tdTomato/UAS-GCaMP6s; VT57358-Gal4/UAS-GCaMP6s |
| 3 A-E bottom and G | TPN3 GCaMP | UAS-cd8tdTomato/UAS-GCaMP6s; R11H09-Gal4/UAS-GCaMP6s |
| 3 – fs1 A | TPN1-GCaMP sugar-TRP | Gr5a-LexA/X; UAS-GCaMP5/UAS-cd8tdTomato;<br>R30A08-Gal4/LexAop-dTRP |
| | TPN1-GCaMP bitter-TRP | Gr66a-LexA/X; UAS-GCaMP5/UAS-cd8tdTomato;<br>R30A08-Gal4 /LexAop-dTRP |
| 3 – fs1 B | TPN2-GCaMP sugar-TRP | Gr5a-LexA/X; UAS-GCaMP5/UAS-cd8tdTomato;<br>VT57358-Gal4 /LexAop-dTRP |
| | TPN2-GCaMP bitter-TRP | Gr66a-LexA/X; UAS-GCaMP5/UAS-cd8tdTomato;<br>VT57358-Gal4 /LexAop-dTRP |
| 3 – fs1 C | TPN3-GCaMP sugar-TRP | Gr5a-LexA/X; UAS-GCaMP5/UAS-cd8tdTomato;<br>C220-Gal4/LexAop-dTRP |
| | TPN3-GCaMP bitter-TRP | Gr66a-LexA/X; UAS-GCaMP5/UAS-cd8tdTomato;<br>C220-Gal4/LexAop-dTRP |
| 4A | TPN2 & MB | MB-dsRed/UAS-GFP; VT57358-Gal4/ TM2 or 6b |
| 4F | TPN3 & MB | MB-dsRed/UAS-GFP; R11H09-Gal4/TM2 or 6b |
| 4 B-E | TPN2 & olfactory PN | GH146-QUAS, QUAS-mtdTomato / UAS-GFP; VT57358-Gal4/+ |
| 4 G-J | TPN3 & olfactory PN | GH146-QUAS, QUAS-mtdTomato / UAS-GFP; R11H09-Gal4/+ |
| 5 A & B | PER: noGal4 | UAS-CsChRimson/X; +; + |

*Table 1 continued on next page*

*Table 1 continued*

| Figure | Description | Genotype |
|---|---|---|
| 5 A & B | PER: GR Gal4s | UAS-CsChRimson/X; Gr64f-Gal4/+; TM2 or TM6b/+ <br> UAS-CsChRimson/X; Gr66a-Gal4/+; TM2 or TM6b/+ |
| 5 A & B | PER: TPN Gal4s | UAS-CsChRimson/X; +; R30A08-Gal4/+ <br> UAS-CsChRimson/X; +; VT57358-Gal4/+ <br> UAS-CsChRimson/X; +; R11H09-Gal4/+ |
| 5 C & D | Shi PER behavior | + / UAS- Shibire(ts) <br> R30A08-Gal4/UAS-Shibire(ts) <br> VT57358-Gal4/UAS-Shibire(ts) <br> R30A08-Gal4, VT57358-Gal4/UAS-Shibire(ts) <br> R11H09-Gal4/UAS-Shibire(ts) <br> C220-Gal4/UAS-Shibire(ts) |
| 5 – fs1A & B | Gr-Gal4s | Gr64f-Gal4/+; TM2 or TM6b/+ <br> Gr66a-Gal4/+; TM2 or TM6b/+ |
| 5 – fs1A & B | TPN-Gal4s | R30A08-Gal4/+ <br> VT57358-Gal4/+ <br> R11H09-Gal4/+ <br> C220-Gal4/+ |
| 5 – fs1 C & D | TPN-Gal4s (controls for PER with Shi) | R30A08-Gal4/+ <br> VT57358-Gal4/+ <br> R11H09-Gal4/+ <br> C220-Gal4/+ |
| 6B | Memory – TPN1 Shi | R30A08-Gal4/UAS-Shibire(ts) |
| 6C | Memory – TPN2 Shi | VT57358-Gal4/UAS-Shibire(ts) |
| 6D | Memory – TPN3 Shi | R11H09-Gal4/UAS-Shibire(ts) |
| 6F | Memory – TPN3 Chrimson | UAS-CsChRimson/X; +; R11H09-Gal4/+ |
| 6 – fs1A | TPN3 – 2nd Gal4 | X; +; C220-Gal4/UAS-Shibire(ts) |
| 6 – fs1B | TPN3 – 2nd Gal4 | UAS-CsChRimson/X; +; C220-Gal4/+ |
| 7 A & B | Memory – TPN2 Shi | VT57358-Gal4/UAS-Shibire(ts) |
| 7 C & D | Memory – TPN3 Shi | R11H09-Gal4/UAS-Shibire(ts) |
| 8 & 9 | PPL1 - control | UAS-CsChRimson/X; LexAop-GCaMP6s; TH-LexA/TM3-ser |
| 8 & 9 | PPL1 – TPN2 | UAS-CsChRimson/X; LexAop-GCaMP6s; TH-LexA/ VT57358-Gal4 |
| 8 & 9 | PPL1 – TPN3 | UAS-CsChRimson/X; LexAop-GCaMP6s; TH-LexA/ R11H09-Gal4 |
| 8 – fs1A | PPL1 imaging | UAS-CsChRimson/X; LexAop-GCaMP6s; TH-LexA/C220-Gal4 |
| 8 – fs1B | PPL1 imaging | UAS-CsChRimson/X; LexAop-GCaMP6s; TH-LexA/TM3-ser <br> UAS-CsChRimson/X; LexAop-GCaMP6s; TH-LexA/C220-Gal4 |
| 8 – fs1C-E | PPL1 ex vivo imaging | UAS-CsChRimson/X; LexAop-GCaMP6s; TH-LexA/ VT57358-Gal4 <br> UAS-CsChRimson/X; LexAop-GCaMP6s; TH-LexA/ R11H09-Gal4 |
| 10 | | Gr66a-LexA, LexAop-CsChrimson / UAS-CD8-tdTomato; UAS-GCaMP6s / MB065B-Gal4 AD; R11H09-Gal4, UAS-GCaMP6s / MB065B-Gal4 DBD |

taste-responsive, we expressed the genetically encoded calcium indicator GCaMP6s (*Chen et al., 2013*) in TPNs to monitor calcium signals (*Figure 3A*). A red indicator was included to assist in locating TPN fibers (*Figure 3B*). TPN activity was monitored in live flies upon stimulation with sucrose, water, or a mixture of bitter compounds applied to the legs (*Marella et al., 2006*). Both TPN1 and TPN2 showed robust responses to sucrose presentation, but did not respond to water or a mixture of bitter compounds. TPN3 responded to the bitter mixture, but not sucrose or water (*Figure 3C–E*). To further confirm that the TPN responses to taste compounds were due to activation of specific gustatory neuron classes, we expressed the heat-activated cation channel dTRPA1 (*Hamada et al., 2008*) in sugar or bitter GRNs, stimulated sensory organs with a local heat probe, and monitored TPN activity with GCaMP6s. Consistent with the responses to taste compounds, TPN1 and TPN2 responded to dTRPA1-mediated activation of sugar GRNs and TPN3 responded to activation of

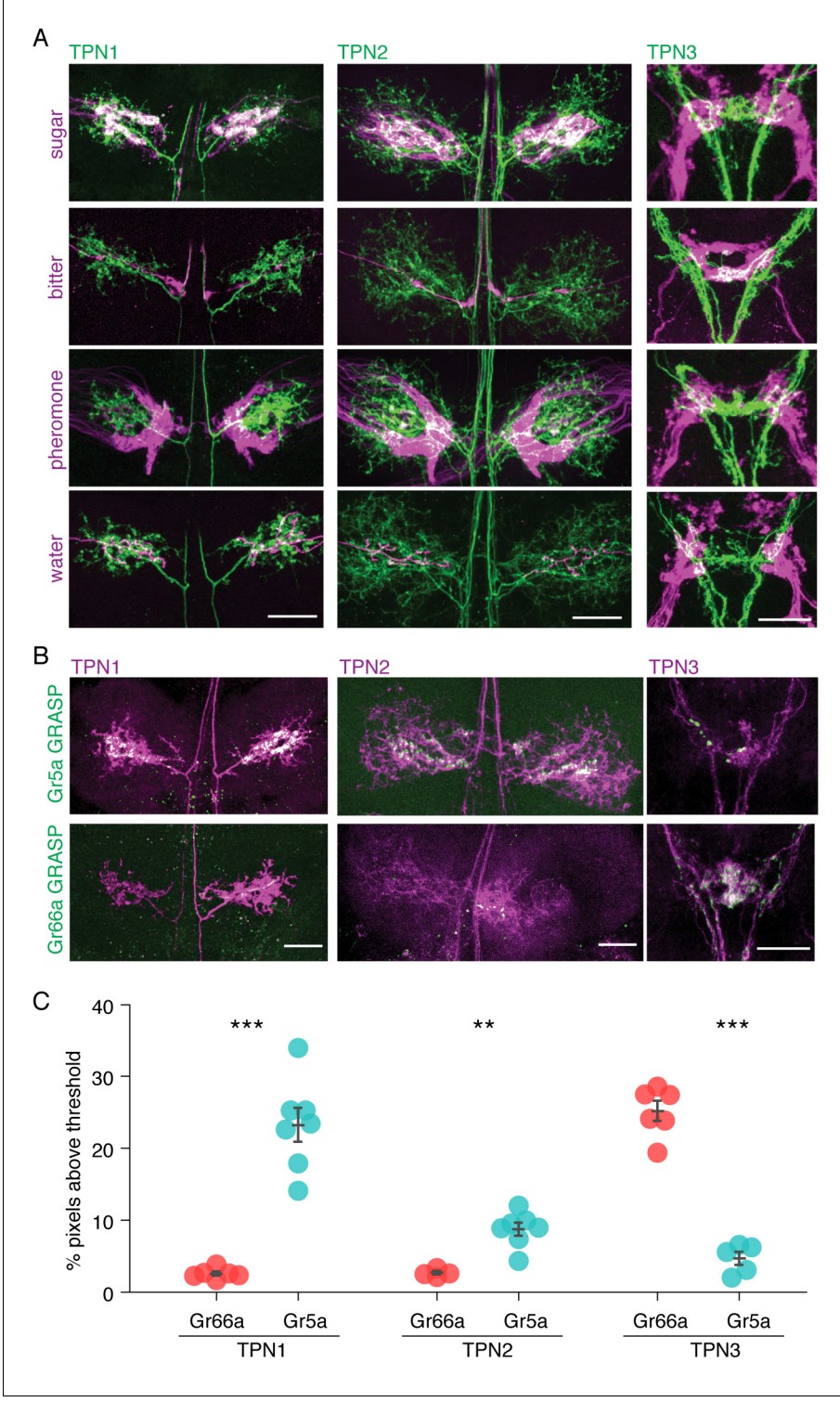

**Figure 2.** TPNs are in close proximity to gustatory projections. (**A**) Double labeling of TPNs (green) with sugar, bitter, pheromone, or water GRNs (magenta). Shown are the projections of TPN1 and TPN2 in the first leg ganglia of the VNC and the projections of TPN3 in the SEZ. TPN1 and TPN2 fibers show strong overlap with sugar but not other GRNs in the VNC. TPN3 shows strong overlap with bitter but not other GRNs in the SEZ. Images shown are z-stacks of the entire preparation and single-plane images are shown in *Figure 2—figure supplement 1*. Scale bar, 25 µm. (**B**) Gr5a (sugar GRN) and Gr66a (bitter GRN) GRASP with TPNs. TPN1 and TPN2 show strong GRASP (green) with sugar GRNs in the VNC while TPN3 shows strong GRASP with bitter GRNs in the SEZ. TPNs are

*Figure 2 continued on next page*

*Figure 2 continued*

labeled with cd8-tdTomato (magenta). Scale bar, 25 μm. (C) Quantification of GRASP (green) signal within the dendritic arbors of the TPNs. Significantly more pixels exceed threshold for sugar (Gr5a) compared to bitter (Gr66a) GRASP for both TPN1 and TPN2. Conversely, significantly more pixels exceed threshold for bitter GRASP for TPN3. *n* = 4–7. Error bars indicate mean ± SEM. Wilcoxon rank sum test: **p<0.01, ***p<0.005.

The following figure supplement is available for figure 2:

**Figure supplement 1.** Single plane examples of TPNs and gustatory projections.

---

bitter GRNs (*Figure 3—figure supplement 1*). Thus, TPNs show modality-specific responses to taste compounds.

TPN1 and TPN2 dendrites are in close proximity to leg gustatory axons, whereas TPN3 arbors are near both leg and proboscis taste axons. To test whether responses were taste organ-specific, we examined responses to taste compounds delivered to proboscis as compared to legs. For TPN1 and TPN2, sucrose delivered to the legs but not the proboscis activated these neurons (*Figure 3F*).

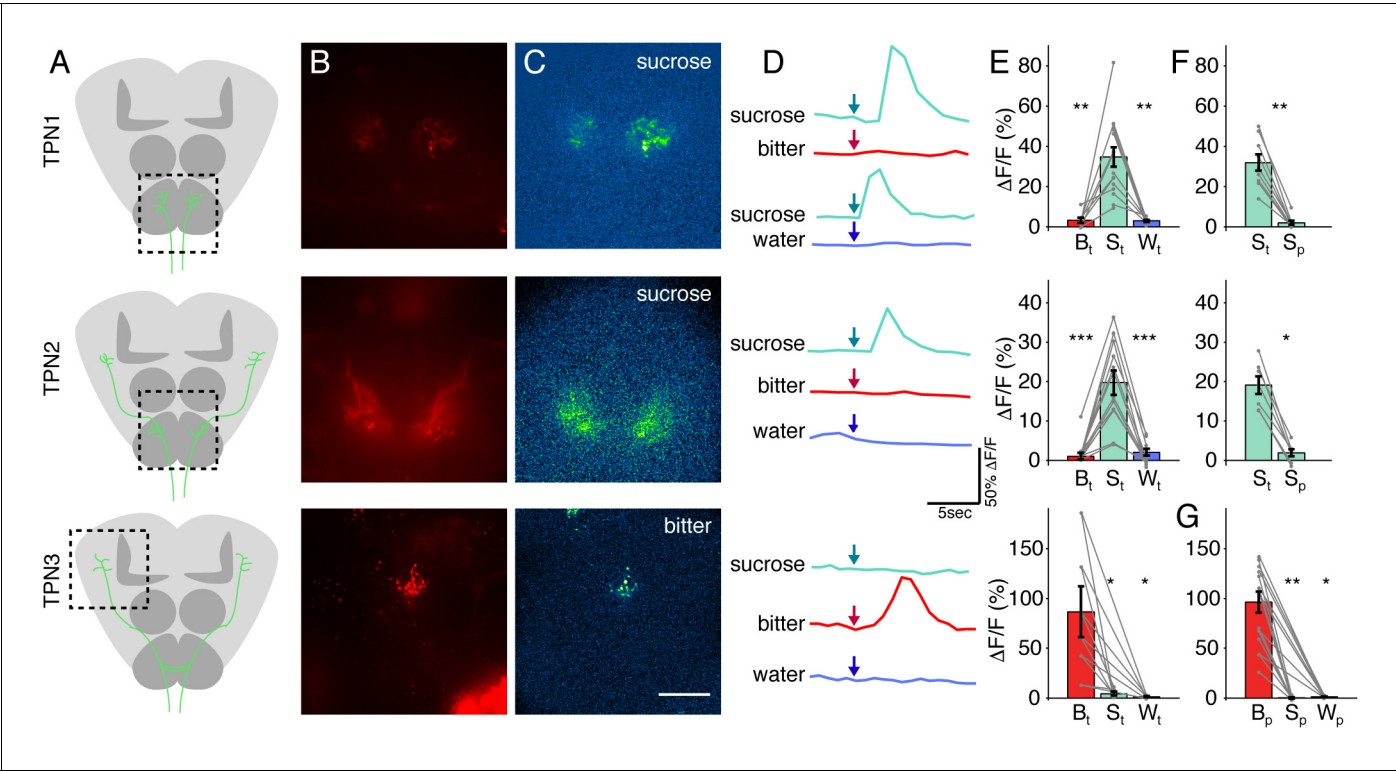

**Figure 3.** TPNs respond to taste compounds. (**A**) Schematic brains showing anatomy of TPNs (green) and approximate region of calcium imaging (dashed box). (**B**) *UAS-cd8-tdTomato* signal used to locate axonal arbors of each neuron type. (**C**) △F signal in response to sucrose (TPN1 and TPN2) or bitter solution (TPN3). Scale bar, 50 μm. (**D**) Example △F/F traces in response to various tastants. Arrows indicate time at which each tastant was presented. (**E**) Summary max △F/F for leg stimulation with each tastant. TPN1 and TPN2 respond only to sucrose, while TPN3 responds only to bitter solution. (**F**) Summary max △F/F for sucrose presentation to legs or proboscis for TPN1 and TPN2. Responses to sucrose are organ specific; both TPN1 and TPN2 only respond to sucrose presentation to the legs. (**G**) Summary max △F/F for proboscis stimulation of each tastant for TPN3. TPN3 shows strong selectivity for bitter presentation to the proboscis as well. For E-G, S: sucrose; B: bitter; W: water; t: tarsi (legs); p: proboscis. Error bars indicate mean +/-± SEM. *n* = 6–12. Paired Wilcoxon tests: *p<0.05, **p<0.01, ***p<0.001.

The following figure supplement is available for figure 3:

**Figure supplement 1.** Calcium Imaging of TPNs with ectopic activation of GRNs.

TPN3 responded to a bitter mixture applied to either the legs or the proboscis (*Figure 3G*). These experiments show that in addition to modality-specific responses to taste compounds, TPNs also show organ-selective responses, consistent with their anatomical arborization patterns.

## TPNs project to Superior Lateral Protocerebrum, near the Lateral Horn

To examine how taste signals from TPNs are transmitted in the brain, we next characterized their axonal arborizations. TPN1 axons terminate specifically in the SEZ, TPN2 has axonal termini in the SEZ and higher brain, and TPN3 axons terminate exclusively in the higher brain (*Figure 1—figure supplement 1B,E,I*). Two of the most studied structures in *Drosophila* higher brain include the mushroom bodies (MB) and lateral horn (LH), regions where second-order olfactory projection neurons terminate (*Marin et al., 2002*; *Wong et al., 2002*). The MBs are essential for learned olfactory associations, whereas the LH has been implicated in innate olfactory behaviors (*de Belle and Heisenberg, 1994*; *Heimbeck et al., 2001*; *Kido and Ito, 2002*). We used double-labeling experiments to assess the relationship between the TPNs, MBs, LH, and olfactory projection neurons (*Figure 4*). Both TPN2 and TPN3 axons run parallel to the olfactory projection neuron lateral antennal lobe tract (lALT) (*Figure 4D,I*). TPN2 and TPN3 axons do not contact the MB (*Figure 4A,F*), but terminate predominately in the superior lateral protocerebrum (SLP), with a few synapses in and around the LH in close proximity to olfactory projection neuron axons (*Figure 4B,G*). TPN2 has axons terminating anterior to the LH and along the ventral and medial edges of the LH (*Figure 4C*). TPN3 also has axons terminating anterior to the LH, along the medial edge of the LH (*Figure 4H*). Thus, TPN2 and TPN3 project near olfactory projection neurons, following along the lALT, with some terminal fibers in the LH but most in the SLP. The SLP is a convergence zone that provides inputs and outputs to the *Drosophila* mushroom bodies (*Aso et al., 2014*; *Ito et al., 1998*). This suggests that TPN2 and TPN3 may participate in learned associations.

## TPNs influence but are not required for proboscis extension

TPN1 conveys taste information from the legs exclusively to the SEZ, whereas TPN2 and TPN3 project to the SLP, suggesting different taste-processing streams. We hypothesized that TPN1 might participate in innate feeding behaviors, as motor neurons that drive proboscis extension and ingestion localize to the SEZ (*Gordon and Scott, 2009*; *Manzo et al., 2012*). As TPN2 and TPN3 arborize in a mushroom body association area, we hypothesized that they might participate in learned behaviors. We therefore sought to test the role of TPNs in mediating innate and learned gustatory behaviors.

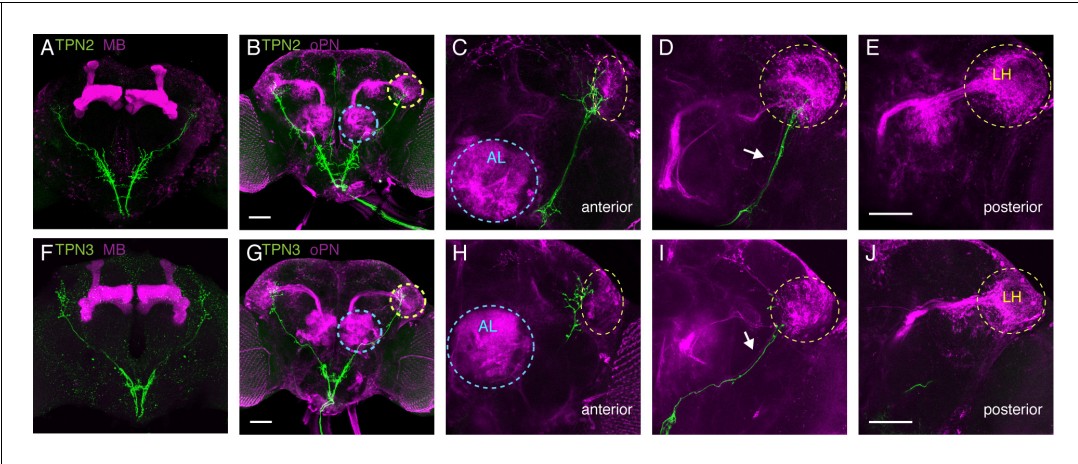

**Figure 4.** TPNs arborize near olfactory projection neurons. (**A**) Co-labeling of TPN2 (green) and mushroom bodies (magenta) shows no overlap. (**B**) Co-labeling of TPN2 (green) and olfactory projection neurons (magenta) shows overlap in lateral horn (LH). Blue circles indicate antennal lobes (AL), yellow circles indicate LH. (**C–E**) Anterior to posterior sections of (**B**). White arrow points to lALT, running parallel to the TPN2 axon. (**F**) Co-labeling of TPN3 (green) and mushroom bodies (magenta) shows no overlap. (**G**) Co-labeling of TPN3 (green) and olfactory projection neurons (magenta) shows overlap in LH. (**H–J**) Anterior to posterior sections of (**G**). Scale bar, 50 μm.

The proboscis extension response (PER) is an innate gustatory-driven behavior: when gustatory neurons on the legs detect sugar, the fly extends its proboscis to eat, and extension is inhibited upon inclusion of bitter compounds (*Dethier, 1976*). To examine whether TPN activation might influence this innate behavior, we selectively activated TPNs with the red-light-gated cation channel CsChrimson (*Klapoetke et al., 2014*) and examined the effects on PER. In the absence of a taste stimulus, activation of either TPN1 or TPN2 increased PER, whereas activation of TPN3 did not elicit PER (*Figure 5A* and *Figure 5—figure supplement 1A*). To test whether activation of TPNs modulates PER in the presence of a moderately appetitive gustatory stimulus, we presented flies with sucrose on their legs while activating TPNs. Activation of TPN1 or TPN2 did not significantly increase PER in this context. However, simultaneous activation of TPN3 with sucrose presentation significantly decreased PER (*Figure 5B* and *Figure 5—figure supplement 1B*). As activation of sugar-sensing TPN1 and TPN2 promotes PER whereas activation of bitter-sensing TPN3 inhibits PER, these experiments demonstrate that neurons that project to the SEZ (TPN1 and TPN2) as well as those that project to protocerebrum (TPN2 and TPN3) can influence SEZ sensorimotor circuits involved in proboscis extension.

Protocerebral pathways may feed back onto the SEZ and serve a modulatory rather than primary function in PER circuits or they might represent essential neurons for this behavior. To determine if TPNs are necessary for PER, we transiently blocked their synaptic output using the temperature-sensitive, dominant-negative dynamin allele, Shibire^ts (Shi^ts) (*Kitamoto, 2001*) while flies were presented with gustatory stimuli. We found that TPN silencing did not affect PER to a moderately appetitive stimulus (100 mM sucrose) (*Figure 5C* and *Figure 5—figure supplement 1C*). When exposed to a bitter-sweet mixture that evoked low PER in controls (6–11%), flies with synaptic output blocked in TPNs generally showed no change in PER rate (*Figure 5D* and *Figure 5—figure supplement 1D*). Small increases that were observed were not consistent across experiments: while TPN2 silencing showed a modest effect, this was not recapitulated when silencing both TPN1 and TPN2. In addition, only one of the TPN3 driver lines showed a small increase in PER rate. Thus, TPNs are not essential for PER and additional pathways must contribute to this innate behavior.

## TPNs are essential for a learned behavior, conditioned taste aversion

In addition to eliciting innate behaviors, taste plays a key role in learned behavior (*Das et al., 2014*; *Masek and Scott, 2010*). To test whether TPNs participate in learned behaviors, we examined whether they were necessary for conditioned taste aversion, a taste-driven behavior that requires the MBs, a learning center in the fly brain (*Keene and Masek, 2012*; *Kirkhart and Scott, 2015*; *Masek and Scott, 2010*). In conditioned taste aversion, flies are presented with a conditioned stimulus (CS, 500 mM sucrose) on the legs, causing PER, whereupon the bitter unconditioned stimulus (US, 50 mM quinine) is applied briefly to the proboscis. This pairing causes a strong reduction in PER upon subsequent sucrose delivery to the legs that recovers after about 30 min.

Sugar on the legs was paired with bitter delivery to the proboscis for five pairings at two-minute intervals, then memory was assessed by presenting the CS alone and recording PER at five-minute intervals (*Figure 6A*). In control animals, a strong reduction in PER occurred during pairing that persisted for several minutes in response to the CS alone (*Figure 6B–D*, gray lines). Blocking synaptic transmission in TPN1 throughout the entire experiment with Shi^ts did not affect aversive taste conditioning (*Figure 6B*), arguing that TPN1, which terminates in the SEZ, is not required for conditioned taste aversion. Surprisingly, inhibiting either sugar-responsive TPN2 or bitter-responsive TPN3 throughout the experiment led to a substantial loss of conditioned taste aversion (*Figure 6C,D* and *Figure 6—figure supplement 1A*). Although TPNs are not required for innate proboscis extension responses to sugar or bitter compounds, the TPNs projecting to the protocerebrum are essential for conditioned taste aversion.

## Manipulating TPN activity during training or testing suggests that TPN3 conveys the bitter US signal and TPN2 conveys the sugar CS

Because TPN3 responds to bitter taste compounds, it is most likely to convey the bitter US in conditioned taste aversion, whereas TPN2 responds to sugar and may convey the CS. To begin to test whether TPN3 might act as the bitter US, we tested whether CsChrimson-mediated activation of TPN3 could replace the bitter stimulus during training (*Figure 6E*). Indeed, flies presented with

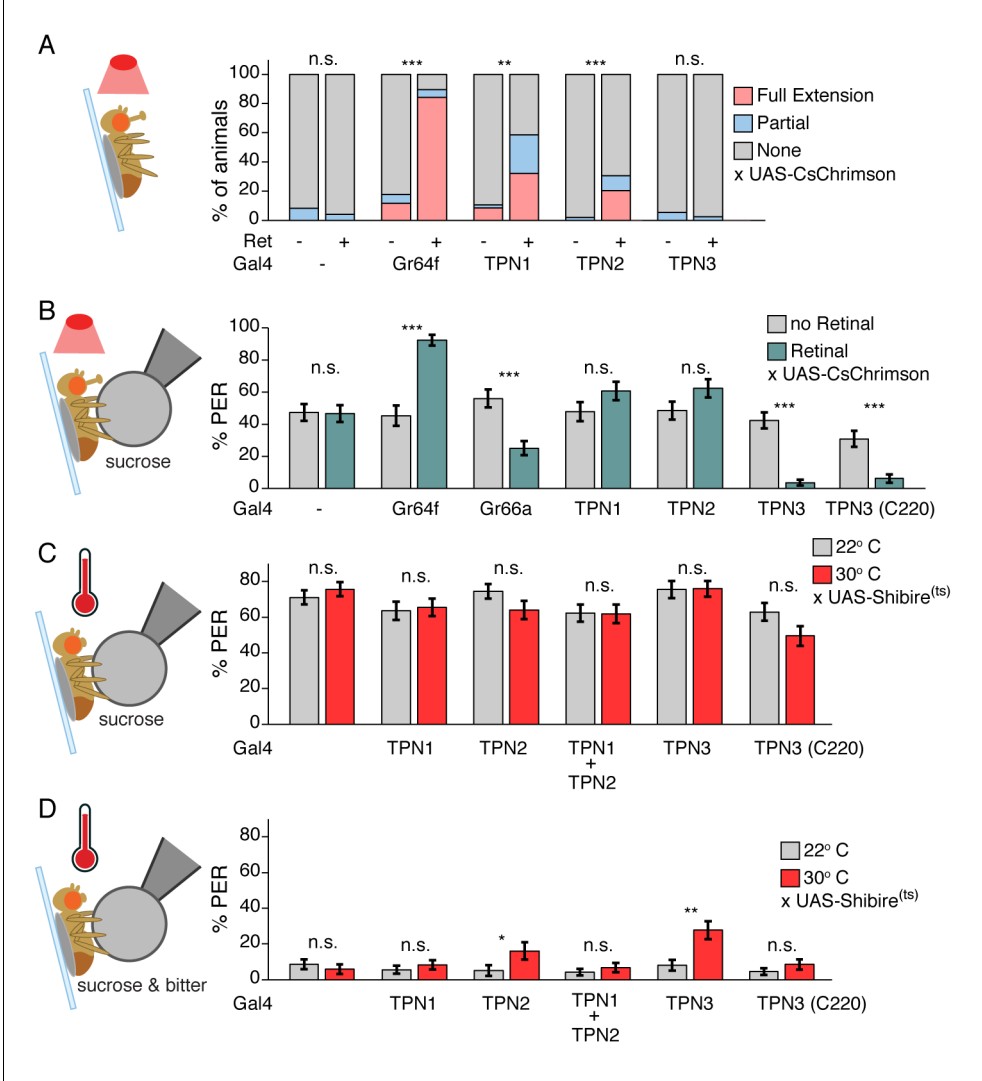

**Figure 5.** Taste projection neurons influence proboscis extension. (**A**) PER to activation with CsChrimson, in absence of a taste stimulus. Activation of sugar GRNs (Gr64f), TPN1, and TPN2 increase PER, compared to no retinal controls. For Gal4 control and Gr64f-Gal4, n = 17–24. For TPNs, n = 38–53. Fisher's exact tests, **p<0.01, ***p<0.001. (**B**) PER to simultaneous CsChrimson activation and 100 mM sucrose presentation. Strong enhancement of PER was observed upon sugar GRN activation, but not TPN1 or TPN2 activation. Strong suppression of PER was observed for both bitter GRN (Gr66a) and TPN3 activation. PER suppression by TPN3 was observed with both Gal4 drivers. n = 42–58. Error bars indicate mean +/-± SEM. Wilcoxon rank-sum tests, ***p<0.001. (**C**) TPNs were conditionally silenced with Shi[ts] and PER was tested to a sweet solution (100 mM sucrose). Silencing TPNs did not affect PER to sweet solutions. n = 50–63. Error bars indicate mean +/-± SEM. Paired Wilcoxon tests. (**D**) TPNs were conditionally silenced with Shi[ts] and PER was tested to a sweet-bitter mixture (100 mM sucrose, 25 mM caffeine, and 0.5 mM denatonium). Small increases in PER rate in some experiments were not consistently found across conditions. n = 50–63. Error bars indicate mean +/-± SEM. Paired Wilcoxon tests, *p<0.05, **p<0.01.

The following figure supplement is available for figure 5:

**Figure supplement 1.** Genetic controls for PER experiments.

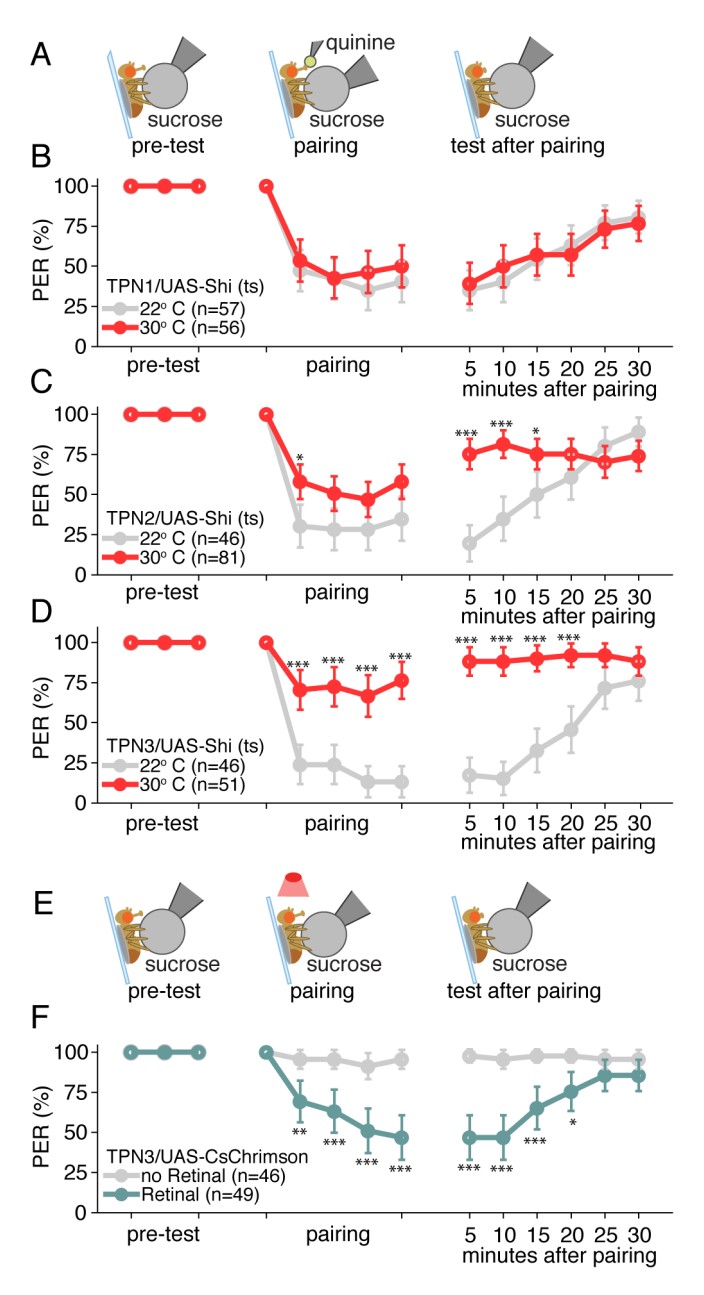

**Figure 6.** Taste projection neurons are essential for conditioned taste aversion. (**A**) Schematic of aversive taste memory protocol experiments with flies expressing Shi[ts] in TPN1, TPN2 or TPN3. Flies were initially presented with 500 mM sucrose (pre-test) and only flies that showed reliable PER were included for the remainder of the experiment. Next, flies were presented with sucrose on their legs and their proboscis was touched with bitter 50 mM quinine upon PER (pairing). Finally, flies were presented with sucrose to the legs in 5 min intervals to test memory after pairing. (**B**) Aversive taste memory upon conditional silencing of TPN1 shows unimpaired memory at restrictive (30°C) temperature (red) when compared to permissive room temperature flies (gray), indicating TPN1 is not required for memory. (**C**) Conditional silencing of TPN2 shows impaired memory at restrictive temperature, specifically in tests after pairing, suggesting TPN2 is required for memory. (**D**) Conditional silencing of TPN3 produces defects during training and post-training, showing TPN3 is required for memory. (**E**) Schematic of aversive taste memory protocol experiments with flies expressing CsChrimson in TPN3. Experiments were conducted as described in A, except during pairing: proboscis stimulation with quinine was replaced with red light stimulation (i.e. activation of TPN3). (**F**) Activating TPN3 with CsChrimson, which replaces bitter application during training, serves as a US to produce robust learning. For B,C,D,F, error bars are 95% CI; for each CS presentation,

*Figure 6 continued on next page*

*Figure 6 continued*

the difference between the control and experimental conditions (e.g. gray and colored data points) were tested with Fisher's Exact Tests with Bonferroni adjusted alpha levels of *p<0.005 (0.05/10), **p<0.001 (0.01/10), ***p<0.0001 (0.001/10).

The following figure supplement is available for figure 6:

**Figure supplement 1.** Memory experiment controls.

sugar as the CS and red-light mediated activation of TPN3 as the US retained an aversive memory that persisted for at least 15 min. This demonstrates that ectopic TPN3 activation can serve as a US (*Figure 6F* and *Figure 6—figure supplement 1B*), indicating that TPN3 conveys aversive punishment signals to higher brain centers involved in this behavior.

To further test TPN3's role as the US conduit, we selectively silenced TPN3 during either the pairing phase or the testing/retrieval phase of the memory protocol (*Figure 7C,D*). Since the US is only presented in the pairing phase, we would predict that TPN3 would be required only during pairing and not during retrieval. Indeed, silencing TPN3 during the pairing phase substantially disrupted memory formation while silencing during the testing phase had no effect. This further supports the notion that TPN3 carries the US signal.

As TPN2 responds to sugar, we hypothesized that TPN2 carries the sugar CS signal in our learning paradigm. As the sugar CS is present during both training and testing, silencing TPN2 during either the pairing phase or the testing phase might be predicted to affect conditioned taste aversion. Consistent with this notion, silencing TPN2 during testing or training led to strongly impaired conditioned taste aversion (*Figure 7A,B*). This demonstrates that TPN2 is required for both the formation and the retrieval of the memory and argues that TPN2 carries the CS signal.

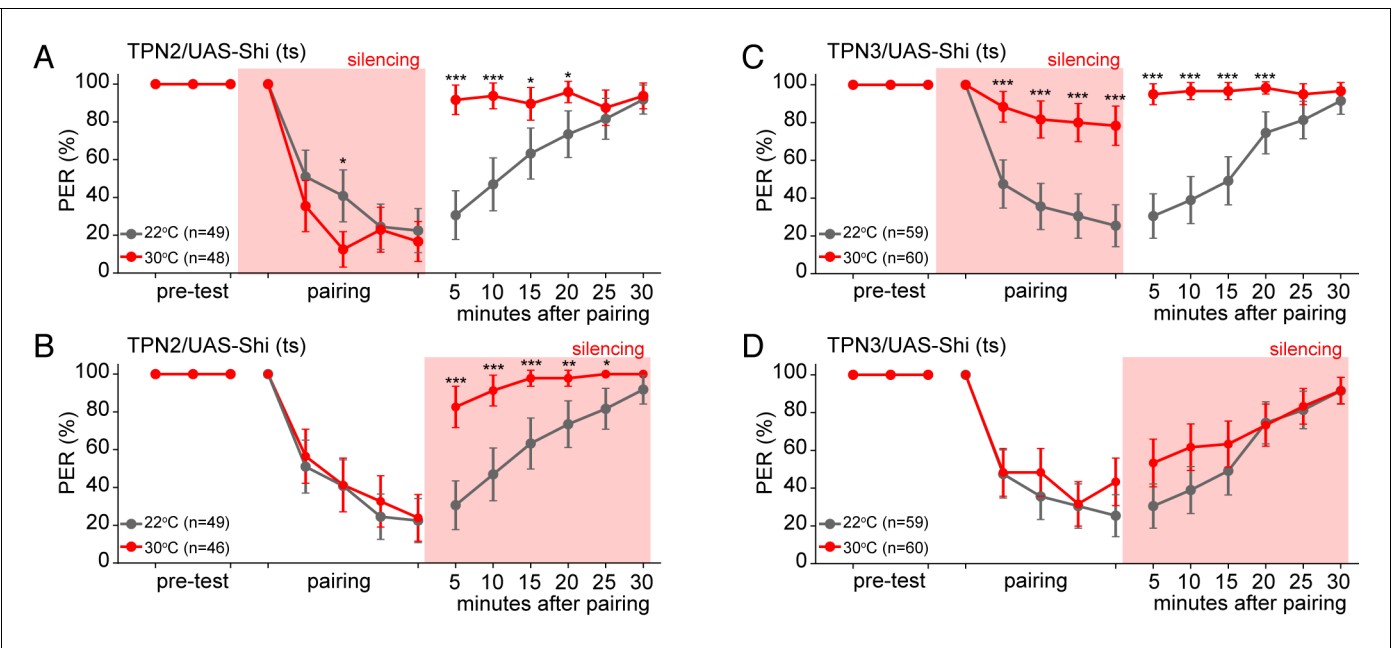

**Figure 7.** TPN2 carries the CS signal, while TPN3 carries the US signal. A and B. Aversive taste memory is impaired upon conditional silencing of TPN2 during either the pairing (A) or the testing/retrieval (B) phases, suggesting TPN2 carries the CS signal. (C) Conditional silencing of TPN3 during the pairing phase shows the inability to form a memory, suggesting TPN3 carries the US signal. (D) Conditional silencing of TPN3 during the testing phase has no impact on memory. Error bars are 95% CI; Fisher's Exact Tests with Bonferroni adjusted alpha levels of *p<0.005 (0.05/10), **p<0.001 (0.01/10), ***p<0.0001 (0.001/10).

## TPNs impinge on the MB learning circuit

Conditioned taste aversion, like many learned associations in the fly, requires the MBs, the site where multiple sensory cues are integrated. The principal cells of the MB are the 2000 Kenyon cells (KCs), which receive olfactory, gustatory, and visual cues as conditioned stimuli (CS) at their dendrites in the calyx region (*Davis, 2005*; *Kirkhart and Scott, 2015*; *Turner et al., 2008*; *Vogt et al., 2016*). KC axons receive reward and punishment signals (US) transmitted via aminergic neurons that segment the MB lobes (*Burke et al., 2012*; *Claridge-Chang et al., 2009*; *Liu et al., 2012*; *Mao and Davis, 2009*). Pairing of the US with the CS causes a synaptic change such that the CS is now sufficient to drive the US behavior (*Hige et al., 2015*). In the conditioned taste aversion paradigm, the bitter US is likely conveyed by PPL1 dopaminergic neurons, as these neurons encode punishment signals, respond to bitter sensory stimulation, and are required for conditioned taste aversion. Similar to olfactory CS, the sugar CS is likely conveyed via KC dendrites, as tastes activate KC dendrites with sparse modality-specific representations (*Kirkhart and Scott, 2015*). PAM dopaminergic neurons that respond to sugars and encode reward are not required for conditioned taste aversion (*Burke et al., 2012*; *Das et al., 2014*; *Kirkhart and Scott, 2015*; *Liu et al., 2012*).

In principle, TPNs might influence learned behaviors by acting upstream of mushroom body learning centers (by providing a US or a CS signal to the MBs), by acting downstream of MB circuits (by providing a copy of taste information to the SLP that is then modified by MB outputs), or both. To test whether TPNs act upstream of MB circuits, we examined whether TPN3 activation can elicit responses in MB extrinsic neurons, specifically the dopaminergic PPL1 cluster which encodes the bitter US (*Kirkhart and Scott, 2015*; *Masek et al., 2015*), although TPN3 and PPL1 are not directly connected (data not shown). To examine a functional link, we monitored GCaMP6s activity in PPL1 neurons while using red light to activate CsChrimson expressed in TPN3 (*Figure 8A*). Red light activation of TPN3 drove strong calcium responses in most PPL1 neurons (*Figure 8B–D*, *Figure 8—figure supplement 1A,B*, *Figure 9*), demonstrating that TPN3 provides an excitatory drive onto PPL1. This excitatory drive is apparent even in an ex vivo preparation (*Figure 8—figure supplement 1C–E*), excluding the possibility that responses are due to fluctuating oscillations seen in vivo (*Cohn et al., 2015*). These results argue that TPN3 activates dopaminergic PPL1 neurons to convey the bitter US signal to the MBs.

If TPN2 acts as the sugar CS, the prediction would be that TPN2 would elicit activity in the MB calyx, as this is the site that receives olfactory CS signals as well as by taste compounds. We activated TPN2 with CsChrimson while monitoring GCaMP6s activity in the MB calyx but were unable to reliably detect activation (data not shown). As another strategy to ask whether TPN2 could act upstream of MB circuits, we asked whether TPN2 influenced activity in aminergic inputs into MBs. Recent studies revealed that activity in dopaminergic MB inputs, including PPL1, is oppositely regulated by appetitive and aversive stimuli (*Cohn et al., 2015*). We therefore monitored activity in PPL1 neurons while activating TPN2 with CsChrimson and found that TPN2 activation significantly decreased PPL1 calcium levels (*Figure 8B–D* and *Figure 9*). The TPN2-dependent changes in PPL1 activity does not readily fit mushroom body learning models is which sugar is the CS signal, and further investigation is required to determine the relevance for conditioned taste behavior. Nevertheless, these findings demonstrate that PPL1 activity is oppositely regulated by TPN2 and TPN3 and argue that both TPN2 and TPN3 may act upstream of the mushroom bodies.

We have shown that TPN3 is required for conditioned taste aversion and that TPN3 positively feeds onto the dopaminergic PPL1 neurons. Does TPN3 exclusively carry the bitter US information to PPL1 neurons? To test this, we asked whether bitter stimuli no longer activated PPL1 when TPN3 was ablated. We generated flies in which bitter GRNs express CsChrimson, allowing for exogenous activation with red light, and monitored calcium signals in both TPN3 and PPL1-MV1 neurons (*Figure 10A,B*). A red indicator was also expressed, allowing for the visualization and laser-ablation of TPN3 axons along the lALT tract. Pre-nerve resection, activation of bitter GRNs activated both TPN3 and PPL1-MV1. Post-nerve resection, red-light activation of bitter GRNs no longer elicited calcium responses from TPN3 (*Figure 10C,G*). In addition, responses in PPL1-MV1 neurons were significantly reduced in magnitude (*Figure 10D,H*). Importantly, these results are not due to non-specific effects of laser treatment, as 'mock' laser ablations to the antennal lobe did not cause significant changes in calcium signals for either TPN3 or PPL1-MV1 (*Figure 10B,E,F,I,J*).

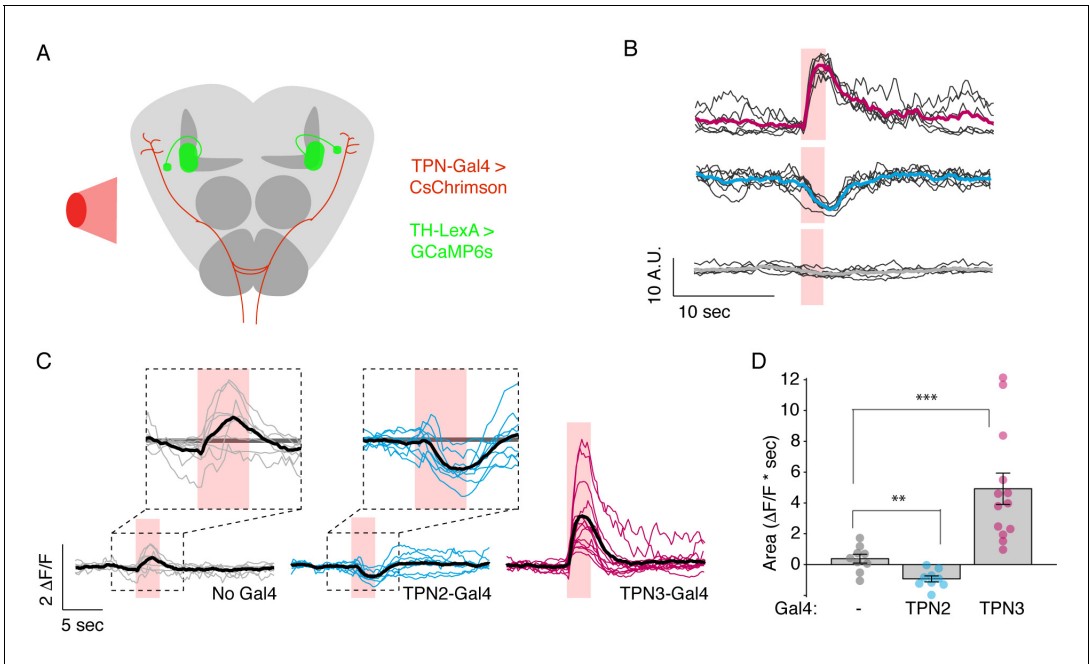

**Figure 8.** TPNs act upstream of mushroom bodies to influence learned behaviors. (**A**) Schematic for imaging calcium responses of PPL1-MV1 neurons to CsChrimson activation of TPN2 and TPN3. (**B**) Example single-animal light-triggered averages of PPL1-MV1 calcium responses to red light activation of TPN3 (top trace), TPN2 (middle trace), and control (bottom trace). Pink background indicates 3 s of red-light stimulation. Individual trails are shown with thin black lines, average trace with thick colored lines. TPN3 activation causes robust increases in PPL1 activity, while TPN2 activation causes a decrease in PPL1 activity. (**C**) Average PPL1-MV1 calcium responses across multiple animals. Thin colored traces are PPL1-MV1 recordings from individual animals, thick black traces are averages. (**D**) Average area under the curve. Compared to no Gal4 controls, TPN2 and TPN3 show significant decreases and increases in calcium signal, respectively. Error bars indicate mean ± SEM. $n$ = 9–13. Wilcoxon rank-sum: **p<0.01, ***p<0.001.

The following figure supplement is available for figure 8:

**Figure supplement 1.** Controls for PPL1 imaging experiments.

We conclude that TPN3 carries bitter US signals to the PPL1 cluster, but additional pathways must exist.

## Discussion

The gustatory system plays a critical role in survival, guiding animals to accept caloric food and avoid toxins. Although the molecular and functional identity of taste receptor neurons in the periphery has been established, the pathways that process gustatory information in higher brain have not been elucidated. Here, we identify three novel classes of taste projection neurons in *Drosophila melanogaster* that convey taste detection from the periphery to higher brain centers. Two classes are sugar-responsive and their activation promotes PER, an innate feeding behavior, whereas one class is bitter-responsive and its activation inhibits PER. Conditional silencing of taste projection neurons has little effect on PER, suggesting parallel pathways, but dramatically influences a learned behavior, conditioned taste aversion. The neurons essential for conditioned taste aversion terminate in the superior lateral protocerebrum and act upstream of mushroom body learning centers. These studies demonstrate modality-selective taste pathways to higher brain, show that taste pathways to the SLP are required for learned associations but not for innate proboscis extension, and suggest similarities in the routing of sensory information in the fly brain.

### Labeled lines for taste processing

Large-scale calcium-imaging studies of taste processing in mammals argue for a labeled line model of taste coding, in which different taste modalities are processed in segregated streams

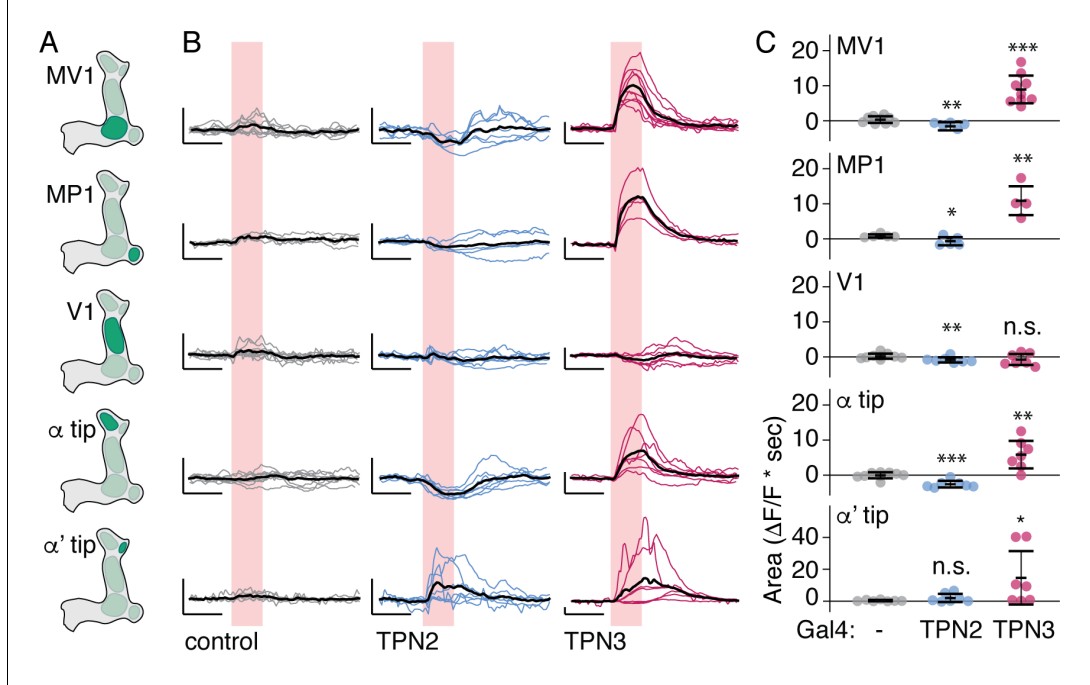

**Figure 9.** TPNs action on all PPL1 lobes. (**A**) Schematic specifying PPL1 region focused on for calcium imaging, with CsChrimson activation of TPN2 or TPN3. (**B**) Average response (black line) and individual traces (gray, blue, and red lines) of PPL1 regions to red light (no Gal4 driver control, TPN2 activation, and TPN3 activation, respectively). Control animals show no responses to red light, while activation of TPN3 generally increases calcium signals in all PPL1, and activation of TPN2 generally results in small but reliable decreases in calcium signal. Pink bar indicates time of laser stimulation. Horizontal scale bar = 5 s. Vertical scale bar = 1 △F/F. (**C**) Average area under the curve. Compared to no Gal4 controls, TPN2 and TPN3 show significant decreases and increases in calcium signal, respectively. Error bars indicate mean +/-± SEM. $n$ = 9–13. Wilcoxon rank-sum: *p<0.05, **p<0.01, ***p<0.001.

(*Chen et al., 2011*). Similar large-scale calcium-imaging studies of taste processing in *Drosophila* suggest that labeled-line encoding of tastes is likely to be an ancient strategy shared across evolution (*Harris et al., 2015*). The existence of separate channels for different tastes suggests a strategy to ensure innate responses to critical compounds.

The identification of specific neurons that process gustatory cues is essential to further test models of modality specificity. Recent studies in *Drosophila* have identified interneurons as well as motor neurons that respond selectively to sugars (*Flood et al., 2013*; *Gordon and Scott, 2009*; *Kain and Dahanukar, 2015*; *Yapici et al., 2016*). Our work extends these studies by identifying long-range projection neurons that separately carry sweet or bitter information to higher brain, demonstrating modality-specific relays. Moreover, TPN1 and TPN2 selectively relay sugar taste detection from the legs, with unilateral or bilateral leg sensory inputs and unilateral axonal projections. This organotopy suggests that taste detection from different organs serves different functions, consistent with studies identifying interneurons that sense sweet taste from the mouthparts and drive ingestion (*Yapici et al., 2016*). In addition, the lateralized sugar projections that we identified may allow sugar detection to be finely localized to the left or right legs, as previously described (*Kirkhart and Scott, 2015*). In contrast, TPN3 responds to bitter taste on the legs and the proboscis, suggesting that aversion to bitter compounds may not require pinpointing location. The organ-specific and modality-specific connectivity of TPNs demonstrates a mechanism to encode taste location in addition to taste quality.

## Functional separation of taste pathways for innate and learned responses

Feeding initiation begins with the proboscis extension response (PER), a rapid and innate behavior that occurs upon appetitive taste detection. Sensory (*Wang et al., 2004*), motor (*Gordon and Scott,*

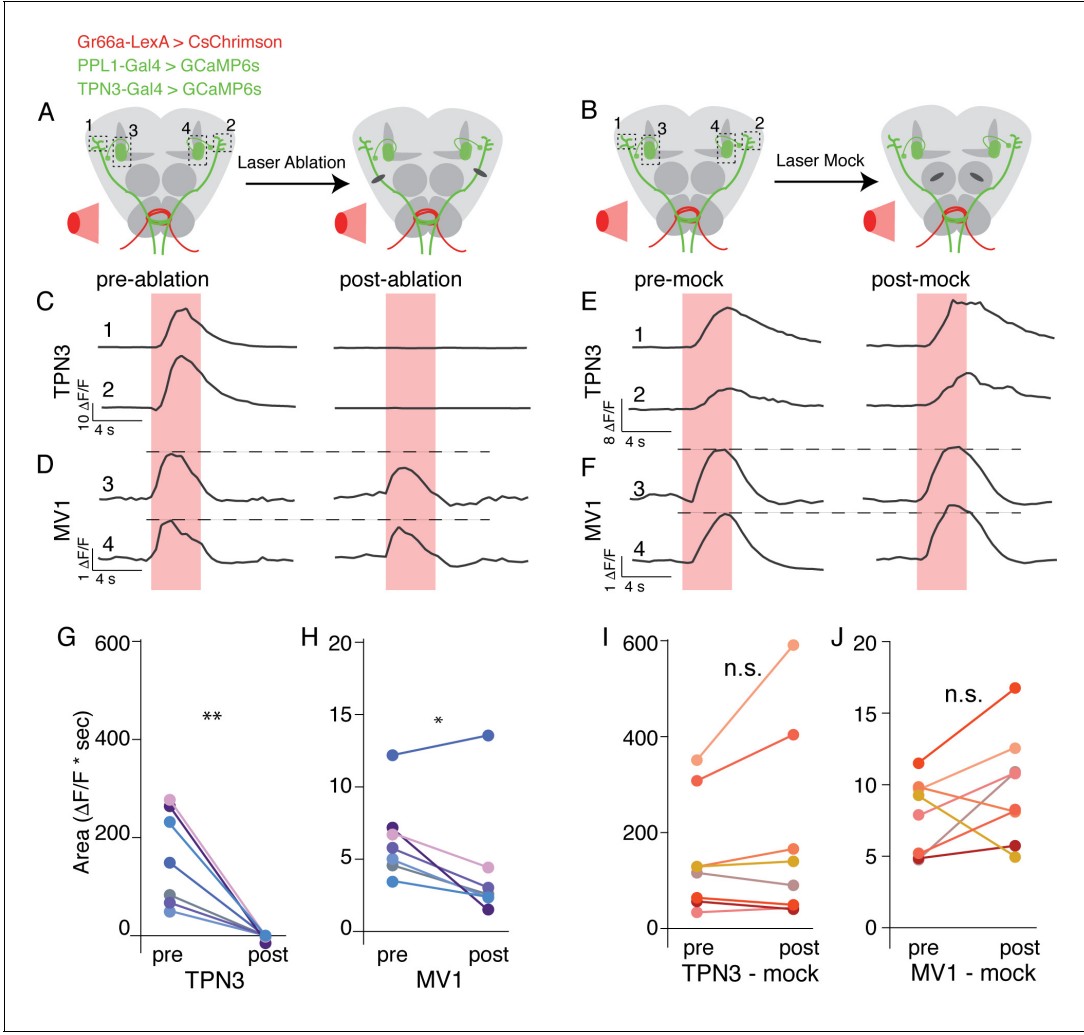

**Figure 10.** TPN3 is not the only relay for bitter information to mushroom bodies . (**A** and **B**) Schematic of imaging and nerve-cut experiment. Bitter GRNs are activated via CsChrimson and calcium responses of TPN3 and the PPL1-MV1 cluster are imaged. Dashed boxes indicate approximate imaging window. (**A**) TPN3 axon is cut with multi-photon laser ablation and resection is verified with calcium imaging. (**B**) The multi-photon laser ablation is directed at the antennal lobe for a "mock ablation." (**C**) Single animal example traces of calcium responses of TPN3 to red light stimulation of bitter GRN. Strong TPN3 response of red light pre-ablation (left) is completely abolished with axon resection of TPN3 (right). (**D**) Single animal example trace of calcium responses of PPL1-MV1 to red light stimulation of bitter GRN. Responses in PPL1-MV1 persist, but are reduced after TPN3 axon resection. Same animal as C. Horizontal dashed line indicates peak of pre-ablation response. (**C** and **D**) Trace numbers correspond to dashed boxes in A. Pink region indicates time of light stimulation. (**E** and **F**) Single animal example traces of calcium responses of TPN3 (**E**) and PPL1-MV1 (**F**) to red light stimulation of bitter GRN, before and after mock ablation. Responses of TPN3 and PPL1-MV1 do not change substantially after mock ablation. Trace numbers correspond to dashed boxes in B. Pink region indicates time of light stimulation. (**G** and **H**) Summary quantification of data with area under the curve for before and after TPN3 nerve cut. All animals showed complete loss of calcium response in TPN3 neurons (**G**) after laser ablation of nerve. Most animals showed an attenuation of response in PPL1-MV1 cluster after TPN3 laser ablation (**H**). n = 7. Wilcoxon rank-sum: *p<0.05, **p<0.01. Matched colors in **G** and **H** indicate the same animal. (**I** and **J**) Summary quantification of data before and after mock ablation. No significant change was observed in calcium response in TPN3 neurons (I) or PPL1-MV1 cluster (J) after mock ablation. n = 8. Wilcoxon rank-sum: n.s. = not significant. Matched colors in I & J indicate the same animal.

2009; Manzo et al., 2012), and modulatory (Mann et al., 2013; Marella et al., 2012) neurons for PER all converge in the SEZ, suggesting that local circuits drive this behavior. Here, we show that activating sugar-sensing TPN1 or TPN2 promotes PER, whereas activating bitter TPN3 inhibits PER. As both TPN2 and TPN3 send axons to the SLP, this suggests that information from the higher brain feeds back onto sensorimotor circuits for PER.

These TPNs could be primary components of taste circuits or may play modulatory roles without being essential components of PER circuits themselves. Conditional inhibition of synaptic transmission in TPNs had little or no effect on proboscis extension (*Figure 5*). These data argue that TPNs are not essential for proboscis extension and other neurons must contribute to this behavior.

Although they are not required for proboscis extension, TPN2 and TPN3 are essential for conditioned taste aversion (*Figures 6* and *7*). Inhibiting synaptic transmission in sugar-sensing TPN2 during either training or testing decreased conditioned aversion, whereas inhibiting bitter TPN3 decreased aversion only if inhibition occurred during training. These behavioral results are most consistent with the notion that TPN2 carries the CS signal and TPN3 encodes the bitter US. The notion that TPN3 carries the bitter US is further supported by behavioral experiments showing that TPN3 activation is sufficient to replace bitter as the US and by calcium imaging experiments showing that TPN3 activates the PPL1 US punishment neurons that wrap around MB lobes (*Figures 8* and *9*). Our observation that bitter sensory neurons activate PPL1 when TPN3 is ablated argues that there are additional bitter pathways that lead to PPL1 (*Figure 10*). The requirement of TPN3 for conditioned taste aversion but not for PPL1 bitter activation is difficult to reconcile without greater understanding of the underlying pathways for the behavior.

How does sweet-sensing TPN2 contribute to conditioned aversion? Innate responses to sugar are unimpaired upon TPN2 silencing, whereas conditioned responses are impacted. If TPN2 conveys the sugar CS signal, we would predict that TPN2 transmits information to Kenyon cell dendrites similar to olfactory CS, but we were unable to observe this. Nevertheless, our calcium-imaging data show that TPN2 negatively regulates PPL1 activity, arguing that TPN2 can provide an input into MB circuits, although the significance for conditioned taste aversion behavior is not clear. An additional possibility is that TPN2 might provide a copy of sugar acceptance to the SLP that is then modified downstream of MB learning circuits. Although our studies clearly show that TPNs influence activity in MB inputs, they do not exclude the possibility that TPNs also send signals to additional neurons that act upstream or downstream of MB circuits.

Here, we show that TPN2 and TPN3, which arborize in the SLP and lateral horn, excite or inhibit MB extrinsic neurons, providing direct evidence for a functional link from TPNs to MB. This is consistent with the view that reciprocal and bidirectional interactions between the SLP and MBs are critical for learned associations (*Aso et al., 2014*).

## Similarities in the logic of sensory processing

Studies of olfactory projection neurons set the paradigm for sensory processing in the fly brain, with odor representations in the mushroom bodies and lateral horn suggesting segregation of learned and innate pathways (*Jefferis et al., 2007*; *Marin et al., 2002*; *Wong et al., 2002*). Examination of olfactory projection neurons that convey carbon dioxide detection suggested that innate signals for aversion project to the lateral horn and are conveyed to the MBs for context-dependent associations (*Bräcker et al., 2013*). Recent studies of thermosensory processing revealed that different classes of thermosensory projection neurons synapse in the MBs and others in the SLP (*Frank et al., 2015*; *Liu et al., 2015*). We find that taste projection neurons run parallel to the lateral antennal lobe tract (lALT) similar to a subset of thermosensory and olfactory projection neurons and terminate in the SLP with a few fibers in the lateral horn. The common routing of different sensory modalities to the SLP area suggests that this may be a site of sensory integration or action selection.

Our studies are consistent with the notion that the SLP is a multimodal sensory area that provides input to mushroom body learning centers. Representations in the SLP may also be modified by MB outputs and may feed back onto innate behavioral programs, like proboscis extension. The observation that the routing of gustatory information shares similarities to olfactory and thermosensory pathways suggests that sensory processing for innate and learned behaviors may be similar across sensory modalities.

## Materials and methods

### Fly stocks

Flies were raised on standard fly food, except for experiments involving CsChrimson. Flies were raised at 25°C, except for experiments involving temperature-sensitive Shibire (flies raised at 20°C).

The following fly strains were used. TPN Gal4 drivers include TPN1: *R30A08-Gal4* (*Jenett et al., 2012*); TPN2: *VT57358-Gal4* (Dickson, unpublished); and TPN3: *R11H09-Gal4* (*Jenett et al., 2012*) and *C220-Gal 4* (*Gohl et al., 2011*). *Gr64f-LexA* (*Miyamoto et al., 2012*); *Gr64f-Gal4* (*Dahanukar et al., 2007*); *Gr5a-LexA*, *UAS-CD4::spGFP1–10*, *LexAop-CD4::spGFP11*, *tub>Gal80>* (*Gordon and Scott, 2009*); *Gr66a-LexA*, *ppk28-LexA*, *UAS-cd8-tdTomato* (*Thistle et al., 2012*); *Gr66a-Gal4* (*Scott et al., 2001*); *ppk23-LexA* (gift from B. Dickson); *MB247-dsRed* (*Riemensperger et al., 2005*); *GH146-QF, Q-UAS-tdTomato* (*Potter et al., 2010*); *UAS-CD8::GFP* (*Lee and Luo, 1999*); *LexAop-CD2GFP* (*Lai and Lee, 2006*); *UAS-DenMark* (*Nicolaï et al., 2010*); *UAS-synaptogmin-eGFP* (*Zhang et al., 2002*); *UAS-RedStinger* (*Barolo et al., 2004*); *MKRS, hs-FLP* (Bloomington); *UAS-GCaMP6s* (*Chen et al., 2013*); *UAS-GCaMP5* (*Akerboom et al., 2012*); *LexAop-GCaMP6s* (gift from D. Kim); *LexAop-dTRPA1* (gift from B. Pfeiffer); *UAS-CsChrimson* (*Klapoetke et al., 2014*); *UAS-shibire^ts* (*Kitamoto, 2001*); *TH-LexA* (*Galili et al., 2014*); *MB065B-Gal4AD, MB065B-Gal4DBD* (*Aso et al., 2014*).

## Immunohistochemistry

Antibody staining was performed as previously described (*Wang et al., 2004*). The following primary antibodies were used: rabbit anti-GFP (Invitrogen, Carlsbad, CA 1:1000), mouse anti-GFP (Invitrogen 1:1000), mouse anti-GFP (Sigma-Aldrich, St. Louis, MO 1:200 - for GRASP staining only), chicken anti-GFP (1:500), mouse anti-nc82 (Developmental Studies Hybridoma Bank, Iowa City, IA 1:500), rabbit anti-RFP (Clonetech, Mountain View, CA 1:500). The following secondary antibodies were used (all Invitrogen at 1:100): 488 anti-rabbit, 488 anti-mouse, 488 anti-chicken, 568 anti-rabbit, 568 anti-mouse, 647 anti-mouse. All images were acquired on a Zeiss confocal microscope. Brightness and contrast were adjusted using FIJI. For quantification of GRASP, we used the following steps for each image: (1) We made a mask (M) of the TPN dendritic field using Huang threshold in FIJI; (2) Using the distribution of GFP outside the mask ($M_{out}$), we choose the 99th percentile as the threshold (T); (3) We measured the percent of pixels inside the mask (M) that exceed the threshold (T).

## Calcium imaging and analysis

Flies were tested 4–6 days post-eclosion.

### TPN calcium imaging

Calcium transients were imaged in flies expressing two copies of *UAS-GCaMP6s* and one copy of *UAS-cd8-tdTomato* on a fixed-stage 3i spinning disk confocal microscope with a piezo drive and a 20x water objective (2x optical zoom), as described (*Harris et al., 2015*). Flies were prepared as described for electrophysiology (*Marella et al., 2012*), with the brain immersed in AHL, while the taste organs remained dry and accessible for stimulation with natural taste stimuli.

Flies were stimulated with water, sucrose (500 mM), or bitter mixture (100 mM caffeine and 10 mM denatonium) solutions to either the legs or proboscis, while the SEZ (TPN1 and TPN2) or lateral protocerebrum (TPN3) were imaged. For SEZ imaging (TPN1 and TPN2) the antennae, cuticle, and underlying air sacs were removed. For leg stimulation experiments (*Figure 3E*), the proboscis was removed and all legs remained intact. For experiments with both proboscis and leg stimulations, the proboscis was carefully waxed out and esophagus cut to allow an unobstructed view of the SEZ (*Figure 3F*). For lateral protocerebrum imaging (TPN3), the proboscis was waxed out, the antennae were removed, and the dorsal cuticle and underlying air sacs were removed. For proboscis stimulation experiments, the legs were removed (*Figure 3G*). For all leg stimulation, approximately 10 uL of solution was suspended around a cube of 2% agarose and secured on a micromanipulator. The tastant was manually advanced to allow all six freely moving legs to contact the tastant. The approximate time of stimulation was noted for all experiments. Variability in latency likely arises from the manual presentation and the volitional movement of the fly. For proboscis stimulation, tastants were presented to the proboscis via capillary tube (*Kirkhart and Scott, 2015*). dTRPA1 stimulation during imaging was performed as previously described (*Kirkhart and Scott, 2015*). The timing of calcium signals does not allow us to distinguish between direct and indirect connections.

*UAS-cd8-tdTomato* was used to define the imaging volume and fast z-sectioning with a piezo drive allowed for volumetric 4-D imaging. Anatomy scans were taken with a 561 nm laser periodically throughout the experiment to help correct for movement. Calcium signal was imaged with a 488 nm

laser, and each z-plane scanned in 100msec, such that the interval between each volumetric scans was 0.8 to 1.3 s (interval length depends on number of z sections). For natural tastant experiments, every fly tested was presented with at least two different tastants, or the same tastant to different organs, (to allow within-fly comparisons) and each tastant was presented to the fly at least twice, with careful notation of when the tastant was presented. Any given fly's response to a tastant is an average of all repetitions. For GRN dTRPA1 activation experiments, only one heat stimulation was taken.

The red anatomy scan was used to select z slices for analysis and delineate unbiased region of interests (ROIs) of axonal terminals. For every fly, four to eight z slices with clear anatomy were chosen for analysis such that we could ensure the same z-planes were analyzed across the various tastant stimulations. A max projection across z for the anatomy scan and each GCaMP time point was used for analysis. Anatomical ROIs were drawn by hand and moved as necessary to correct for X-Y movement. In addition, a large ROI was drawn in a region that did not express GCaMP to measure background autofluorescence. Mean fluorescence levels from the background ROI were subtracted from the anatomical ROI at each timepoint, resulting in fluorescence trace over time: F(t). $\triangle$F/F (%) was measured as follows: 100% * (F(t) – F(0)) / F(0). Max $\triangle$F/F was measured by subtracting the average $\triangle$F/F signal from the three timepoints preceding tastant presentation from the maximum $\triangle$ F/F value after tastant presentation. ROI drawing and fluorescence measures were done in FIJI. Analysis was done in Matlab.

## PPL1 cluster calcium imaging

Calcium transients of the PPL1 lobes were imaged in flies expressing two copies of *LexAop-GCaMP6s* driven by *TH-LexA* and one copy of *UAS-CsChrimson* driven by specific Gal4 driver or no driver (control) on a Zeiss LSM 780 NLO AxioExaminer with a 20x water objective. 48 hr prior to calcium imaging, flies were placed in the dark on standard fly food supplemented with 0.4 mM all-trans retinal (Sigma). 24 hr prior to experiments, flies were starved in the dark on a wet kimwipe (0.4 mM all-trans retinal). Calcium signal was imaged with 925 nm 2-photon excitation on a single plane and TPNs were stimulated with a 635 nm red laser (LaserGlow). To capture the various PPL1 lobes, different scans were taken at different z-planes.

Flies were prepared for calcium imaging as described above for lateral protocerebrum imaging. Given the oscillatory nature of PPL1 neurons (*Plaçais et al., 2012*), three to eight red laser pulses were presented in each experiment. For ex-vivo imaging, flies were prepared and stimulated as described above, except the whole central brain and VNC were dissected and pinned to a sylgard dish.

Mean fluorescence levels from a background ROI were subtracted from the PPL1 ROI, resulting in fluorescence trace over time: F(t). $\triangle$F/F was measured as follows: (F(t) – F(0)) / F(0). Area under the curve ($\triangle$F/F * sec) was integrated over five seconds starting from the onset of the red laser. ROI drawing and fluorescence measures were done in Zen or FIJI. Analysis was done in Matlab.

## Multi-photon laser-mediated ablations

Laser ablations and calcium imaging were conducted on a Zeiss LSM 780 NLO AxioExaminer. Animals were prepared and calcium imaging was conducted as described for PPL1 cluster imaging above. Two copies of *UAS-GCaMP6s* and one copy of *UAS-CD8-tdTomato* were driven by *TPN3-Gal4* and a split Gal4 for PPL1. In addition, one copy of *LexAop-CsChrimson* was driven by the bitter GRN driver *GR66a-LexA*. Laser ablation protocol were reported previously (*Kallman et al., 2015*). Briefly, TPN3 axons were visualized using a 561 nm laser and small ROI was drawn on the axon. The ROI was scanned with a multi-photon laser tuned to 760 nm at ~50% laser strength (~50 mW at the front lens) until the axon became discontinuous. Successful resection was physiologically tested with calcium imaging and laser ablation was repeated as necessary.

## Behavioral experiments

Flies were tested 3 to 11 days post-eclosion. For activation experiments, *UAS-CsChrimson* flies were used and neurons were activated with a 635 nm red laser (LaserGlow). 48 hr prior to behavioral testing, experimental *UAS-CsChrimson* flies were placed in the dark on standard fly food with 0.4 mM all-trans retinal (Sigma), while control flies were placed in the dark on standard fly food. 24 hr prior

to behavioral testing, flies were starved in the dark on a wet kimwipe (0.4 mM all-trans retinal for experimental flies). For inactivation experiments, *UAS-Shi^ts* flies were raised in a 20°C incubator. 24 hr prior to behavioral testing, flies were starved on a wet kimwipe. All flies were anesthetized with $CO_2$, glued dorsal side down on a glass slide, and allowed to recover in a humid chamber (in the dark for CsChrimson experiments) for 2–3 hr before behavior tests.

## Proboscis Extension Response (PER)

PER in response to activation of TPNs via CsChrimson was tested in two ways. First, we tested the effect of activating the TPNs in the absence of a tastant. Flies were exposed to 5 s of red light for two trials. Flies were considered to have no extension if the proboscis did not move for both trials, full extension if a full PER was observed in either trial, and a partial extension if the proboscis partially extended for at least one trial. We used a 2 × 3 Fisher's Exact Test to whether the proportions of flies responding to red light was statistically different. Second, we simultaneously activated the TPNs while presenting 100 mM sucrose on the tarsi of the flies. This moderately appetitive stimulus results in proboscis extension 50% of the time in control flies (e.g. gray bars in *Figure 5B*). Flies were water-satiated before the experiment and in between trials, and presented with the tastant and red light for up to five seconds (or until a PER was observed). Flies were given a score of 0 for no extension, 0.5 for partial extension, and one for full extension, and the average was taken across two trials. We used 2-sample Wilcoxon rank-sum tests since comparisons were made between flies with and without retinal exposure. For both CsChrimson PER experiments, trials were separated by at least five minutes, during which flies were placed in a dark, humid chamber.

To test the necessity of taste projection neurons in PER, we expressed Shi^ts in each TPN class. Flies were presented with a sweet solution (100 mM sucrose) and a sweet-bitter mixture (100 mM sucrose, 25 mM caffeine, and 0.5 mM denatonium) at the permissive room temperature (22–23°C) and the restrictive warm temperature (30–32°C). Flies were water-satiated before the experiment and between trials and given at least 10 min on the heat block or at room temperature before PER testing. Each tastant and temperature combination was tested twice in every fly, and the order of presentation was counterbalanced. Flies were scored as described above and we used paired Wilcoxon signed-rank tests since comparisons were made within a fly on and off the heat block.

## Learning assay

Aversive taste memory was tested as previously described (*Kirkhart and Scott, 2015*). Shi^ts and CsChrimson flies were prepared as described above. Flies were water-satiated before the experiment and between each trial. Flies were presented with 500 mM sucrose (CS) on the legs for three trials before pairing. Any fly that did not extend its proboscis for all three trials was removed from the study. CS presentation was then paired with the US of 50 mM quinine on the proboscis (for Shi^ts experiments) or red light (for CsChrimson experiments) for five trials (2 min inter-trial interval). During pairing, the US was only presented when flies performed a PER in response to the CS. Learning was then tested by presenting the CS alone for six trials (5 min inter-trial interval). For the Shi^ts experiments in *Figure 6*, the entire learning assay was performed at room temperature (control) or on a heat block (experimental). For the Shi^ts experiments in *Figure 7*, experimental animals were placed on the heat block during either the pairing sessions or the testing sessions and compared to control animals maintained at room temperature. For CsChrimson experiments, experimental flies were exposed to all-trans retinal while control flies were not. Fisher's Exact Test was used for the last four pairing trials and the six memory trials with a Bonferroni correction for multiple comparisons (adjusting the alpha value by a factor of 10).

## Statistics

### Imaging

For GRASP quantification (*Figure 2*), Wilcoxon rank-sum tests were used since comparisons were made between flies of different genotypes. For TPN calcium imaging with natural stimuli (*Figure 3*) and calcium imaging of PPL1 with nerve ablation (*Figure 10*), paired Wilcoxon tests were used, since comparisons were done within animals. For TPN calcium imaging with ectopic activation of GRN neurons (*Figure 3—figure supplement 1*) and calcium imaging of PPL1 neurons with ectopic

activation of TPNs (*Figure 8*; *Figure 8—figure supplement 1*; *Figure 9*; *Figure 10*) Wilcoxon rank-sum tests were used.

## Behavior

For PER in response to red light (*Figure 5A* and *Figure 5—figure supplement 1A*), Fisher's Exact Test was used in comparing the portions of PER responses in retinal-exposed vs. no-retinal animals. For PER in response to red light and sucrose (*Figure 5B* and *Figure 5—figure supplement 1B*), Wilcoxon rank-sum tests were used to compare the PER rate in retinal-exposed vs. no-retinal animals. For PER experiments with Shi[ts] (*Figure 5CD* and *Figure 5—figure supplement 1C,D*), paired Wilcoxon tests were used since individual animals were tested both at the permissive room temperature (22°) and the restrictive temperature (30°). For the learning assays (*Figure 6 and 7*, and *Figure 6—figure supplement 1*), Fisher's Exact Test was used for the last four pairing trials and the six memory trials with a Bonferroni correction for multiple comparisons (adjusting the alpha value by a factor of 10).

## Acknowledgements

Members of the Scott laboratory provided comments on the manuscript and B Kallman provided assistance with 2-photon imaging. This work was performed in part at the CRL Molecular Imaging Center, supported by NSF DBI-1041078. This work was supported by a grant from NIDCD to KS and NSF GRFP to CK.

## Additional information

### Competing interests

KS: Reviewing editor, *eLife*. The other authors declare that no competing interests exist.

### Funding

| Funder | Grant reference number | Author |
| --- | --- | --- |
| National Institute on Deafness and Other Communication Disorders | DC013280 | Kristin Scott |
| National Science Foundation | | Colleen Kirkhart |

The funders had no role in study design, data collection and interpretation, or the decision to submit the work for publication.

### Author contributions

HK, Conceptualization, Formal analysis, Investigation, Visualization, Methodology, Writing—original draft, Writing—review and editing; CK, Investigation, Methodology; KS, Conceptualization, Supervision, Funding acquisition, Writing—original draft, Writing—review and editing

### Author ORCIDs

Kristin Scott, http://orcid.org/0000-0003-3150-7210

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
