## [Decision Letter]

Thank you for submitting your article "Long-range projection neurons in the taste circuit of *Drosophila*" for consideration by *eLife*. Your article has been reviewed by three peer reviewers, and the evaluation has been overseen by Hugo Bellen as the Reviewing Editor and a Senior Editor. The reviewers have opted to remain anonymous.

The reviewers have discussed the reviews with one another and the Reviewing Editor has drafted this decision to help you prepare a revised submission. The reviewers discussed the reviews and decided that all the points should be addressed. As you will see all are textual changes and hence I anticipate that you can resubmit in less than two months.

*Reviewer #1:*

Understanding the logic by which sensory processing pathways are organized to guide behavior is a central challenge in neuroscience. In fruit flies, work in the olfactory, visual and gustatory pathways have provided a wealth of insights into how the sensory periphery extracts salient cues, and have exploited largely reflexive behaviors to unravel how this information can guide behavior. However, in the gustatory system, our understanding has been centrally limited by the fact that while genetic tools to characterize primary sensory neurons and output motor neurons have been spectacularly successful, efforts to go deeper into the circuit have met limited success.

In this work, Kim and colleagues identify and characterize three different types of second order taste projection neurons. By screening a number of Gal4 collections, they establish genetic access to these neurons, show that they have dendrites in close proximity to primary taste receptors, and use calcium imaging to show that these cells are taste responsive. They then use genetic tools to activate or silence synaptic transmission from these cells, and discover that while they are not required for innate, reflexive responses to bitter and sweet compounds, they do play critical roles in learned preferences.

Overall, the data in this paper are of high quality, and the overall conclusions of this work will be of interest to a broad audience. I am therefore supportive of publication, and have only one point that I would like the authors to respond to, likely in the text of the manuscript. In particular, I am puzzled by the variable, often extended, response latencies of the TPNs. In Figure 3, the authors show stimulus evoked – calcium traces that sometimes only begin to rise above baseline after a period of 1-2 seconds, but can also be "immediate" (compare 3D top sucrose trace with the bottom trace). Conversely, thermal activation of the input neurons invariably produces immediate responses (Figure 3—figure supplement 1), a result more in line with my expectation of the response of a second order neuron. While I could imagine that the time traces of sensory stimulation as marked in Figure 3 might be somewhat less precise than those of thermal activation, could they really be off by something like two seconds, making this a trivial technical issue? Or is it possible that there is something else of interest here, such as the possibility that processing of natural stimuli might be gated in time depending on the specifics of the trial structure (that is, whether the animal was previously exposed to water or bitter)? Or something else?

*Reviewer #2:*

The manuscript by Kim et al. identifies three novel classes of taste projection neurons (TPN1-3) that convey gustatory information from primary gustatory receptor neurons (GRNs) to other processing stations in the brain. The projection neurons were found in the expression patterns of enhancer fusion Gal4 lines, two of which (i.e. the ones for TPN1 and TPN2) are exquisitely selective, strongly expressing Gal4 only in the TPNs. The third class of projection neuron (TPN3) was identified as a pair of neurons common to two lines that otherwise overlapped little. The authors used these Gal4 lines to characterize the anatomy, projections, gustatory selectivity, functions, and functional connectivity of the three classes of TPNs. The analysis is elegantly presented, the data is compelling, and the conclusions are for the most part novel and of high interest. In particular, the finding that the TPN2 and TPN3 projection neurons are required for the learning and contextualization of gustatory information in a conditioned taste aversion paradigm is of great interest and not only sheds light on how gustatory experience is processed to influence future behavior, but also begins to reveal similarities in the processing streams of different sensory modalities.

Subsection “TPNs project to Superior Lateral Protocerebrum, near the Lateral Horn”: The statement that TPN3 axons terminate "exclusively" in the higher brain seems at odds with the data in Figure 1—figure supplement 1 which shows anti-syt staining in the SEZ. Do the TPN3 neurons terminate in the SEZ?

Subsection “TPNs influence but are not required for proboscis extension”, last paragraph: The inconsistency referred to presumably is in the differing results obtained with the two TPN3 lines, one of which shows a significant difference and the other one of which doesn't. This, however, is not exactly clear and it would be helpful to have a more explicit explanation of this inconsistency.

Subsection “TPNs are essential for a learned behavior, conditioned taste aversion”, first paragraph: I'm not a psychologist, but the description of sugar as a conditioned "appetitive" stimulus doesn't seem right here. Things are already confusing because sweet stimuli elicit robust unconditioned responses and sugar would typically be used as a US in classical conditioning experiments. In the taste aversion paradigm, however, sugar does qualify as a CS because of its acquired aversive properties. My impression is that in psych speak it is thus a "conditioned aversive stimulus" rather than a "conditioned appetitive stimulus," as stated. I'm more than willing to be wrong, but it is probably worth checking. Also, this distinction might be more than semantics and may be relevant to the failure to find TPN2-mediated activation of the MB calyx.

Subsection “TPNs are essential for a learned behavior, conditioned taste aversion”, last paragraph: The loss doesn't seem "complete" in that performance doesn't return to pre-test levels. That is, the responses seem significantly different from 100%. "Substantial" would be a better word than complete. (In this vein, I struggled a bit to understand that the p-values in Figure 6 reflected significance tests performed on the data in gray and red for each time point and not differences from 100%. A statement in the figure legend would be helpful.)

Subsection “Manipulating TPN activity during training or testing suggests that TPN3 conveys the bitter US signal and TPN2 conveys the sugar CS”, second paragraph: Again, the term "completely" seems inaccurate, and should be modified.

Table 1: The data shown in the table for Figure 3 seems to actually be for Figure 4. (Note, for example, that Figure 3 legend says that UAS-cd-tdTom was used to locate arbors, but Table 1 says the red label is MB-dsRed.) In fact, the Figure 3 data seems to be missing from the Table and the figure numbers seem to be offset afterwards. This should be fixed.

*Reviewer #3:*

This manuscript describes neurons in the fly that convey gustatory information to the brain to coordinate innate and learned behaviors. The figures are very nice and the story is well reasoned. I do not have major concerns and would support publishing as is.

Summary:

Sensory neurons detect sweet and bitter tastes. This information is conveyed to higher areas of the brain to direct innate and learned behaviors. This study identifies the first "Taste Projection Neurons" (TPNs) that act between the sensory neurons and higher centers.

Candidate TPNs were identified by anatomical criteria: proximity followed by GRASP contacts with gustatory receptor neurons (GRNs) in the legs and proboscis. TPNs 1 and 2 show GCaMP responses to sugar and TPN3 to bitter compounds, and to genetic activation of GRNs.

Tastes can guide innate responses and learned associations. While the same sensory detectors are required for both behaviors, where the information diverges was unknown. Hunger and sweet taste can evoke proboscis extension (PER) while bitter tastes suppress it. Optogenetic activation of TPN1 or 2 triggers PER but blocking their activity does not inhibit innate PER, suggesting functionally redundant circuits. Blocking TPN2 or 3, however, does block a learned behavior, conditioned taste aversion.

The authors show that the TPNs project through the same tract as the olfactory projections to the Lateral Horn and Superior Lateral Protocerebrum. The SLP may have connections to the mushroom bodies, brain areas involved in learned associations. Activation of TPN3, the bitter responder, can evoke responses in the MB output PPL1 dopaminergic neurons, consistent with TPN3 carrying the aversive, unconditioned stimulus, while activation of TPN2, a sweet responder, reduces the PPL1 response.

Interestingly, blocking of TPN3 can affect the conditioned taste aversion behavior but does not fully suppress PPL1 activation in response to bitter taste, suggesting an alternative pathway for the learned response. Also interestingly, the TPN2 down regulation of PPL1 is not the expected mode of action for the sugar-conditioned inputs. There are clearly future experiments to do to explain these observations.

---

## [Author Response]

*Reviewer #1:*

*[…] Overall, the data in this paper are of high quality, and the overall conclusions of this work will be of interest to a broad audience. I am therefore supportive of publication, and have only one point that I would like the authors to respond to, likely in the text of the manuscript. In particular, I am puzzled by the variable, often extended, response latencies of the TPNs. In Figure 3, the authors show stimulus evoked – calcium traces that sometimes only begin to rise above baseline after a period of 1-2 seconds, but can also be "immediate" (compare 3D top sucrose trace with the bottom trace). Conversely, thermal activation of the input neurons invariably produces immediate responses (Figure 3—figure supplement 1), a result more in line with my expectation of the response of a second order neuron. While I could imagine that the time traces of sensory stimulation as marked in Figure 3 might be somewhat less precise than those of thermal activation, could they really be off by something like two seconds, making this a trivial technical issue? Or is it possible that there is something else of interest here, such as the possibility that processing of natural stimuli might be gated in time depending on the specifics of the trial structure (that is, whether the animal was previously exposed to water or bitter)? Or something else?*

We believe the variability in latency (for natural stimuli) is primarily a technical issue. In tarsal (leg) stimulation protocols, the fly’s legs remain free to move and we use a micro-manipulator to manually advance the stimulus while visualizing the stimulus with a USB pen camera. Therefore, tastant presentation time therefore has two major sources of variability: first, the manual nature of our stimulus presentation adds some variability (but only on the order of 100s of ms); second, the fly’s volitional movement adds substantial variability (a fly that happens to be “reaching” can touch the stimulus much sooner than a fly whose legs are more retracted, certainly adding variability on the scale of 1-2 s). Neither sources of variability are present with the thermal activation (which is initiated with a TTL pulse). We have edited the Methods (subsection “TPN Calcium imaging”, second paragraph) to reflect this.

*Reviewer #2:*

*[…] Subsection “TPNs project to Superior Lateral Protocerebrum, near the Lateral Horn”: The statement that TPN3 axons terminate "exclusively" in the higher brain seems at odds with the data in Figure 1—figure supplement 1 which shows anti-syt staining in the SEZ. Do the TPN3 neurons terminate in the SEZ?*

The anti-GFP staining in Figure 1—figure supplement 1 in the SEZ is due to aberrant expression in the UAS- Synaptotagmin-eGFP line used in this experiment. The green staining in the SEZ is strictly anterior (close to the esophageal opening) and posterior (along the posterior edge of the SEZ) to TPN3’s SEZ arbors. Such aberrant staining is also apparent in Figure 1—figure supplement 1 and Figure 1—figure supplement 1. We have added a line stating this in Figure 1—figure supplement 1 legend.

*Subsection “TPNs influence but are not required for proboscis extension”, last paragraph: The inconsistency referred to presumably is in the differing results obtained with the two TPN3 lines, one of which shows a significant difference and the other one of which doesn't. This, however, is not exactly clear and it would be helpful to have a more explicit explanation of this inconsistency.*

We have edited the results to be more explicit about the inconsistency.

*Subsection “TPNs are essential for a learned behavior, conditioned taste aversion”, first paragraph: I'm not a psychologist, but the description of sugar as a conditioned "appetitive" stimulus doesn't seem right here. Things are already confusing because sweet stimuli elicit robust unconditioned responses and sugar would typically be used as a US in classical conditioning experiments. In the taste aversion paradigm, however, sugar does qualify as a CS because of its acquired aversive properties. My impression is that in psych speak it is thus a "conditioned aversive stimulus" rather than a "conditioned appetitive stimulus," as stated. I'm more than willing to be wrong, but it is probably worth checking. Also, this distinction might be more than semantics and may be relevant to the failure to find TPN2-mediated activation of the MB calyx.*

Agreed, we removed the word “appetitive” to eliminate any confusion.

*Subsection “TPNs are essential for a learned behavior, conditioned taste aversion”, last paragraph: The loss doesn't seem "complete" in that performance doesn't return to pre-test levels. That is, the responses seem significantly different from 100%. "Substantial" would be a better word than complete. (In this vein, I struggled a bit to understand that the p-values in Figure 6 reflected significance tests performed on the data in gray and red for each time point and not differences from 100%. A statement in the figure legend would be helpful.)*

*Subsection “Manipulating TPN activity during training or testing suggests that TPN3 conveys the bitter US signal and TPN2 conveys the sugar CS”, second paragraph: Again, the term "completely" seems inaccurate, and should be modified.*

We changed “completely” to “substantially” for both instances. The figure legend has been modified to be more explicit about the statistical test.

*Table 1: The data shown in the table for Figure 3 seems to actually be for Figure 4. (Note, for example, that Figure 3 legend says that UAS-cd-tdTom was used to locate arbors, but Table 1 says the red label is MB-dsRed.) In fact, the Figure 3 data seems to be missing from the Table and the figure numbers seem to be offset afterwards. This should be fixed.*

Thank you! Table has been corrected.

*Reviewer #3:*

*[…] Interestingly, blocking of TPN3 can affect the conditioned taste aversion behavior but does not fully suppress PPL1 activation in response to bitter taste, suggesting an alternative pathway for the learned response. Also interestingly, the TPN2 down regulation of PPL1 is not the expected mode of action for the sugar-conditioned inputs. There are clearly future experiments to do to explain these observations.*

We thank the reviewer for their positive assessment.